# Both Local Validity and Global Effectiveness Matter: Decoupled Credit Assignment for Long-Horizon Agentic Learning

## Abstract

The natural-language action space of Large Language Model (LLM) agents creates a real risk of invalid outputs (e.g., API rejections, parsing errors). Consequently, in Reinforcement Learning (RL) for long-horizon LLM agents, learning to generate a locally valid action in each turn is as crucial as selecting a globally effective one. However, this requirement was overlooked by the prevailing additive paradigm for credit assignment in agentic RL. Specifically, it computes an action's credit by summing an estimated local score and the trajectory-level score. This paradigm assigns a "contribution" score to all actions regardless of their validity, allowing invalid actions to be assigned positive credit, especially in positive trajectories. To address this, we propose Multiplicative Gated Rewards (MGR), which decouples local action-level validity from global effectiveness. MGR uses a fact-based validity signal, derived from direct environment feedback and syntactic validity, to determine the action-level score (e.g., $\pm 1$). This score is then multiplied by the magnitude of the trajectory-level score. This ensures the action's validity strictly governs the reward's polarity, preventing credit misassignment. Experiments demonstrate that our method improves training stability and achieves SOTA performance on long-horizon LLM agent benchmarks. Code of MGR has been uploaded in the Supplementary Material.

## 1 Introduction

A key frontier for Large Language Model (LLM) agents lies in their ability to solve long-horizon tasks (Yao et al., 2022), often requiring a long chain of interactions to navigate complex environments (e.g., over 10 interactions). Reinforcement Learning (RL) has emerged as a standard for training these agents, enabling them to learn from environmental feedback. Unlike traditional RL agents with discrete action sets, language-based agents face a unique challenge in their vast, language-based action space. They generate actions as text sequences (e.g., code, API calls), creating a fundamental dual requirement for every action: it must be both **locally valid** (e.g., syntactically correct) to be executable, and **globally effective** to contribute to the final task goal. As emphasized by (Zhao et al., 2025), the validity of each turn's action is crucial for long-horizon agents.

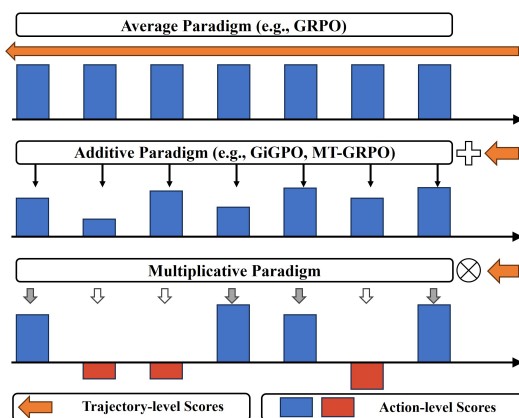

Figure 1: Illustration of Different Credit Assignment Paradigms in a Successful Trajectory.

However, we observe that this essential requirement for local action validity has been largely overlooked by the prevailing credit assignment methods in agentic RL, which we term the *additive paradigm* (as illustrated in Fig. 1). Whether through learned progress estimators (Wu et al., 2025), reward decomposition (Zheng et al., 2025), or sub-

trajectory (Feng et al., 2025) comparisons, this paradigm computes an action's credit by summing an estimated local score with the global score of its parent trajectory. This design is blind to intrinsic validity, and its core flaw is assigning a "contribution" score to all actions regardless of whether they are executable or not. This leads to severe credit misassignment: an unequivocally invalid action may receive a positive reward simply by appearing in a successful trajectory (shown in the second row of Figure 1). Such flawed signals fundamentally destabilize the learning process, forcing the agent to expend valuable samples learning basic syntax from ambiguous feedback, a problem that contributes to training inefficiencies.

To address this, we propose **Multiplicative Gated Rewards (MGR)**, a novel credit assignment framework that decouples the reward signal into two orthogonal components. The first is an Action-Level Sign Provider, which determines the atomic, factual correctness (a positive/negative sign) of an action. This sign is derived from direct environmental feedback and refined with heuristics that model temporal dynamics (e.g., error correction) and penalize unproductive behaviors (e.g., repetition). The second is a Trajectory-Level Magnitude Provider, which employs group-relative methods to evaluate the quality of an entire trajectory, providing the trajectory-level scores.

These two components are fused via a multiplicative gating mechanism, where the final reward is the product of the local signal and the global magnitude. This design ensures that an action's factual validity strictly governs the reward's polarity. It guarantees that invalid actions receive negative rewards even in highly successful trajectories, while valid actions within failed trajectories are selectively preserved as positive learning samples via a dynamic gate. Crucially, this mechanism also gives rise to an implicit curriculum: early in training, the agent is primarily rewarded for mastering foundational skills (i.e., producing valid actions); as it matures, the learning objective implicitly transitions to demanding the strategic composition of these actions to achieve high-quality trajectories. This multiplicative gating mechanism corrects the core flaw of the additive paradigm: an action's factual invalidity can no longer be masked by overall trajectory success. This eliminates a critical vector of credit misassignment by design.

Our contributions are threefold:

- We identify the dual challenge in agentic learning: achieving both local action validity and global strategic effectiveness.

- We propose Multiplicative Gated Rewards (MGR), a new framework that uses a fact-based, action-level signal to gate the credit flowing from a trajectory's global outcome.

- We demonstrate empirically that MGR leads to significant gains in training stability and sample efficiency, achieving SOTA performance on standard agentic benchmarks.

## 2 RELATED WORK

The task of training multi-turn LLM agents has recently seen a surge of interest, with Reinforcement Learning (RL) emerging as a powerful paradigm for enhancing long-horizon decision-making. Our work builds upon and diverges from several key research threads in this area.

### 2.1 TRAJECTORY-LEVEL REINFORCEMENT LEARNING

Initial applications of RL to LLMs often operated at the trajectory level. Methods like Proximal Policy Optimization (PPO) (Schulman et al., 2017) and its variants, including Group-Relative Policy Optimization (GRPO) (Shao et al., 2024), treat an entire sequence of actions as a single data point. The agent receives a sparse reward signal only upon task completion, and this signal is uniformly attributed to all actions within the trajectory. While effective for global policy alignment, this approach suffers from severe credit misassignment noise; it invariably rewards erroneous or suboptimal actions within a successful trajectory and penalizes correct exploratory steps within a failed one. The work by (Shi et al., 2025) systematically identified this issue, noting that without fine-grained, reasoning-oriented feedback, agents can fall into "echo traps" and fail to develop robust, multi-step reasoning.

## 2.2 LEARNED PROCESS REWARD MODELS

Another significant line of work attempts to learn a model to predict turn-level rewards. These are often referred to as Process Reward Models (PRMs) (Gao et al., 2025). For instance, SPA-RL (Chen et al., 2025b) trains a "progress estimator" to redistribute the final task reward back to intermediate steps. Similarly, CAPO (Xie et al., 2025) utilizes a powerful teacher LLM as a generative PRM to produce token-level feedback. These methods replace the sparse reward problem with a new, potentially complex estimation problem. The learned reward, which is a proxy for strategic contribution, is then typically added to the environment's reward stream. This still risks rewarding invalid actions if the estimator model incorrectly infers a positive contribution. Furthermore, they introduce significant computational overhead and a dependency on the quality and potential biases of the reward model itself.

## 2.3 FINE-GRAINED ADVANTAGE ESTIMATION

A third category of methods refines the advantage calculation at a sub-trajectory level, but remains within the foundational additive framework of the Bellman equation. GiGPO (Feng et al., 2025) extends the group-relative concept to the step level by comparing actions from different trajectories at aligned "anchor states." ARPO (Dong et al., 2025) uses model uncertainty (entropy) as a heuristic to identify critical steps and attribute credit accordingly. SPO (Huang et al., 2025) proposes a segment-level advantage estimation as a compromise between turn-level and trajectory-level granularity. While these methods offer more sophisticated estimation techniques, they are still fundamentally estimating a single, conflated score for each step that is implicitly added to form the trajectory's total value. They do not possess a mechanism to strictly enforce the binary validity of an action.

Reinforcement Learning for LLM agents is fundamentally a credit assignment problem. Prevailing methods, which we classify under an *additive paradigm*, often conflate an action's intrinsic validity with its long-term strategic contribution. This leads to credit misassignment, where invalid actions in successful trajectories can erroneously receive positive rewards. To address this, we propose a novel *multiplicative framework* centered on the principle of **signal decoupling**. Instead of learning a single, entangled turn-level advantage, we decompose the learning signal into two orthogonal components: a turn-level sign representing factual validity and a trajectory-level magnitude representing strategic quality.

# 3 METHOD

To decouple reward signals into local and global components, we introduce **MGR (Multiplicative Gated Rewards)**, a multiplicative framework that separates the reward into an action-level signal judging local validity and a trajectory-level magnitude evaluating global strategic contribution. The final reward, $R_{\text{action}}$, is conceptualized as:

$$R_{\text{action}} = \underbrace{\mathcal{G}(R_{\text{local}})}_{\text{Validity Signal}} \times \underbrace{R_{\text{global}}}_{\text{Effectiveness Magnitude}} \tag{1}$$

where $R_{\text{local}}$ is the action-level signal, $R_{\text{global}}$ is the trajectory-level score. And $\mathcal{G}$ is a multiplicative gated module, which fuses the two signals before. This design ensures that credit is assigned correctly: all invalid actions are consistently penalized, while valid actions within failed trajectories can be selectively rewarded to facilitate learning. The algorithm is detailed in Alg. 1.

## 3.1 TRAJECTORY SEGMENTATION AND REINFORCEMENT LEARNING

Our training process begins with collecting a set of $m$ trajectories, $\{\tau_1, \tau_2, \ldots, \tau_m\}$, for each given task. A single trajectory $\tau_i$ of length $T$ is a sequence of states and actions: $\tau_i = (s_1, a_1, s_2, a_2, \ldots, s_T, a_T)$. To form a training batch for our language model-based policy $\pi_\theta$, we segment each trajectory into action-level samples. Each sample $(p'_t, a_t)$ consists of the action $a_t$ and its corresponding prompt $p'_t$, which comprises the entire history up to that point, i.e.,

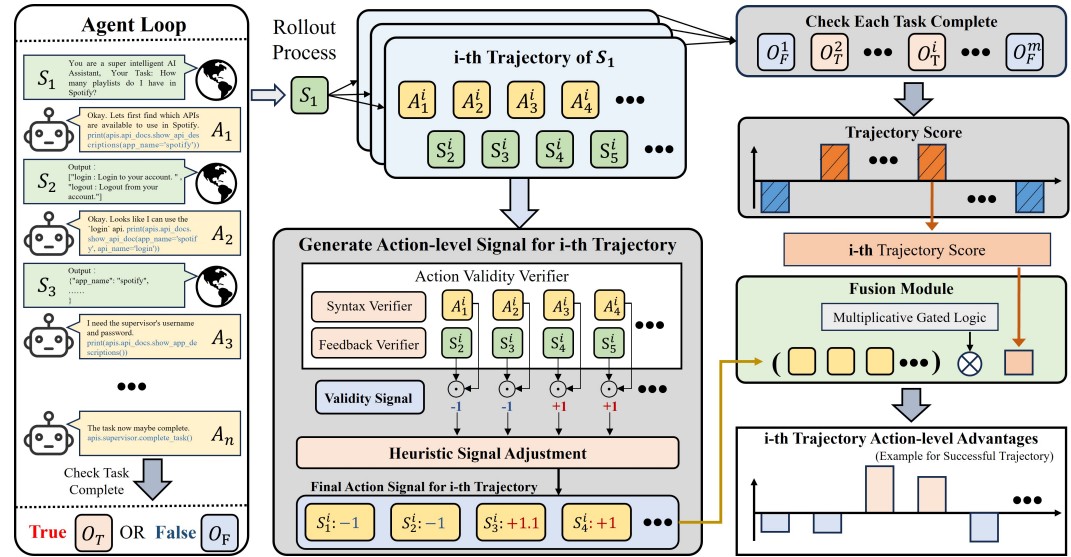

Figure 2: Example of **MGR (Multiplicative Gated Rewards)** calculation in a successful Trajectory. The example in the failure trajectory can be seen in App. A.2.

$p'_t = (s_1, a_1, \ldots, s_t)$. This segmentation process can be visualized as follows:

$$\tau = [s_1, a_1, \ldots, s_T, a_T] \quad \longrightarrow \quad \left\{ \begin{array}{ll} (\text{prompt: } s_1, & \text{response: } a_1) \to r_1 \\ (\text{prompt: } [s_1, a_1, s_2], & \text{response: } a_2) \to r_2 \\ \vdots & \vdots \\ (\text{prompt: } [s_1, a_1, \ldots, s_T], & \text{response: } a_T) \to r_T \end{array} \right\}$$

where each action-level reward $r_t$ is computed by our MGR framework.

Based on MGR's rewards, we then update the policy parameters $\theta$ using the Proximal Policy Optimization (PPO) algorithm. Following prior work (Chen et al., 2025a) that suggests removing the KL-divergence term can be beneficial for agentic learning, we optimize the following objective function:

$$\mathcal{L}^{\text{PPO}}(\theta) = \mathbb{E}_t \left[ \min \left( \rho_t(\theta)\hat{A}_t, \text{clip}(\rho_t(\theta), 1 - \epsilon, 1 + \epsilon)\hat{A}_t \right) \right] \tag{2}$$

where $\rho_t(\theta) = \frac{\pi_\theta(a_t|s'_t)}{\pi_{\theta_{\text{old}}}(a_t|s'_t)}$ is the probability ratio, and $\hat{A}_t$ denotes the token-level advantage in PPO, which is estimated from the MGR rewards $r_t$ via GAE and PPO's critic model. The subsequent sections will provide an exposition of the MGR.

### 3.2 REWARD SIGNAL DECOUPLING

The core principle of MGR is the decomposition of the complex credit assignment problem into two simpler, orthogonal sub-problems, addressed by two distinct signal providers.

#### 3.2.1 ACTION-LEVEL SIGN PROVIDER ($R_{\text{LOCAL}}$)

This component is responsible for determining the local, factual validity of an action $a_t$. It generates a signal $R_{\text{local},t}$, that evaluates the atomic correctness of an agent's action at each turn. The process involves two stages:

**(1) Fundamental Validity Signal ($v_t$):** We first establish a binary signal, $v_t \in \{-1, +1\}$, which serves as a hard constraint on action executability. Let $o_{t+1}$ denote the environment execution feedback following action $a_t$. We define $v_t$ as:

$$v_t = \begin{cases} -1 & \text{if } a_t \notin \Omega_{\text{syntax}} \vee \exists e \in \mathcal{E}_{\text{runtime}}, \text{match}(o_{t+1}, e) \\ +1 & \text{otherwise} \end{cases} \tag{3}$$

where $\Omega_{\text{syntax}}$ denotes the set of actions satisfying predefined structural and formatting constraints (e.g., correct code encapsulation), and $\mathcal{E}_{\text{runtime}}$ represents a set of universal error patterns inherent to the execution environment (e.g., `SyntaxError`, `TimeOut`, or empty responses). The function match($\cdot$) detects if the feedback contains any error pattern $e$. Crucially, this design leverages the standardized Input/Output nature of the most prominent LLM agent domains, such as coding, OS operation, and API interaction. Since these environments universally communicate execution failures via standard protocols (e.g., interpreter stderr, HTTP error codes), our validity check is inherently generalizable across these high-value settings without requiring task-specific state engineering.

**(2) Heuristic Adjustments ($h_t$):** The base signal is then refined with heuristic bonuses and penalties, $h_t$, to capture more nuanced behaviors. Specifically:

- *Heuristic Bonus:* We model the relationship between consecutive turns to encourage recovery from errors: $h_t^{\text{temporal}} = \beta$ if $v_t > 0$ and $v_{t-1} < 0$. Conversely, it receives a penalty for failing immediately after a success: $h_t^{\text{temporal}} = -\beta$ if $v_t < 0$ and $v_{t-1} > 0$.

- *Repetition Penalty:* To prevent reward hacking (e.g., repeatedly using same valid actions) and model stagnation, we apply a penalty for excessive repetition of the same action. This penalty, $h_t^{\text{repeat}}$, is only applied to locally valid actions ($v_t > 0$). Let $N(a_t, \tau)$ be the occurrence count of a specific action $a_t$ among all valid actions within the trajectory $\tau$ up to the current turn. The penalty is activated when $N(a_t, \tau)$ exceeds threshold $q$:

$$h_t^{\text{repeat}} = \begin{cases} -\alpha \cdot (N(a_t, \tau) - q) & \text{if } v_t > 0 \text{ and } N(a_t, \tau) > q \\ 0 & \text{otherwise} \end{cases} \tag{4}$$

   This ensures that once an action is deemed repetitive, each subsequent repetition receives an incrementally larger penalty.

The magnitudes of these heuristic adjustments are controlled by hyperparameters $\beta$ and $\alpha$. Final heuristic adjustments $h_t = h_t^{\text{temporal}} + h_t^{\text{repeat}}$. The final action-level signal of $a_t$ is given by:

$$R_{\text{local},t} = v_t + h_t \tag{5}$$

### 3.2.2 TRAJECTORY-LEVEL MAGNITUDE PROVIDER ($R_{\text{GLOBAL}}$)

This component evaluates the global, strategic quality of an entire trajectory $\tau_i$. For a given task $\tau$, we compare all $m$ collected trajectories. First, each trajectory is assigned a binary score $S(\tau_i)$ based on task completion signal. Following recent advancements in training agents for long-horizon tasks (Chen et al., 2025a), we utilize the Eqn. (6) to transform these absolute scores into a normalized, zero-sum advantage scores. This provides a more stable learning signal by focusing on relative performance. The global magnitude for trajectory $\tau_i$ is calculated as:

$$R_{\text{global}}(\tau_i) = \frac{m}{m-1} \left( S(\tau_i) - \frac{1}{m} \sum_{j=1}^{m} S(\tau_j) \right) \tag{6}$$

### 3.3 REWARD FUSION

The fusion of the local signal $R_{\text{local},t}$ and the global magnitude $R_{\text{global}}$ is performed via a conditional multiplicative process. The core idea is that the final reward's sign should primarily be dictated by the action's local validity ($R_{\text{local}}$), while its magnitude should be scaled by the trajectory's overall effectiveness ($R_{\text{global}}$). The logic is partitioned into three distinct scenarios based on the alignment of the signals' signs:

1. **Concordant signals (sign($R_{\text{local}}$) = sign($R_{\text{global}}$)):** When an action's validity aligns with the trajectory's result (e.g., both the action and the trajectory are either correct or incorrect.), the credit assignment is straightforward multiplication.

2. **Successful trajectory, invalid action** ($R_{\text{global}} > 0, R_{\text{local}} < 0$)**:** The action must be penalized to discourage it, but the penalty is dampened. This is achieved by multiplying the signals and applying a scaling coefficient.

3. **Failed trajectory, valid action** ($R_{\text{global}} < 0, R_{\text{local}} > 0$)**:** This is the most crucial and ambiguous case. We introduce a stochastic gating operator, $g_t$, which determines the sign of the final reward.

This fusion logic is formally captured in the following Equation. Let $\tau(t)$ denote the trajectory containing the action at turn $t$. The final reward $R_{\text{final},t}$ is defined as (more detail in Alg. 1):

$$R_{\text{final},t} = \begin{cases} R_{\text{local},t} \cdot |R_{\text{global}}(\tau(t))| & \text{if } \text{sign}(R_{\text{local},t}) = \text{sign}(R_{\text{global}}(\tau(t))) \\ \gamma \cdot R_{\text{local},t} \cdot R_{\text{global}}(\tau(t)) & \text{if } R_{\text{global}}(\tau(t)) > 0 \text{ and } R_{\text{local},t} < 0 \\ \gamma \cdot g_t \cdot R_{\text{local},t} \cdot |R_{\text{global}}(\tau(t))| & \text{if } R_{\text{global}}(\tau(t)) < 0 \text{ and } R_{\text{local},t} > 0 \end{cases} \tag{7}$$

where $\gamma \in (0, 1]$ is scaling coefficients that reduce the reward magnitude in cases of sign conflict. The key component is the gating operator $g_t \in \{-1, 1\}$, which is sampled based on the agent's overall performance (i.e., task success rate and action accuracy rate). The function of this gate is to dynamically control the balance of positive and negative training samples for locally correct actions, preventing the model from being rewarded for valid but strategically poor choices, especially in later stages of training. The mechanism of $g_t$ is detailed in the next section.

### 3.4 DYNAMIC SAMPLE BALANCING VIA GATED SIGN FLIPPING

As the training progresses, the model rapidly masters the local validity of actions. We observe a shift in the distribution of action-level samples: the ratio of valid to invalid actions transitions from a balanced state (e.g., 40% vs 60%) to a severe imbalance (e.g., 90% vs 10%), even within unsuccessful trajectories.

Without dynamic regulation, this imbalance leads to a critical degradation in the **Signal-to-Noise Ratio (SNR)** of the reward signal. The overwhelming volume of positive rewards for valid actions begins to mask the penalty for the trajectory-level failure, causing the agent to stagnate in local optima (e.g., same valid actions) rather than improving global strategy.

The fundamental issue lies in the intrinsic **ambiguity** of valid actions within failed trajectories. They possess a dual nature depending on the training stage:

- **Early Stage (Signal):** When the agent is weak, these samples are critical *positive signals* for learning basic syntax, preventing policy collapse.

- **Late Stage (Noise):** As validity saturates, these samples become *false positive noise* relative to the strategic objective. High retention of such noise dilutes the gradient direction towards task completion.

To address this, we introduce the gating operator $g_t$ to function as an **implicit curriculum** that manages this SNR transition. By probabilistically flipping the sign of these ambiguous samples, $g_t$ effectively restores the **balance between positive and negative samples**.

For any given failed trajectory $\tau$, a single probabilistic decision is made: either *all* locally valid actions within $\tau$ retain their positive sign, or they are *all* flipped to negative. This approach treats the trajectory's outcome as a holistic strategic failure, ensuring that all contributing actions receive a coherent learning signal.

$$p_{\text{retain}} = f_{\text{schedule}}(C_{\text{batch}}, V_{\text{batch}}) = \begin{cases} 1 & \text{if } V_{\text{batch}} < \theta_V \text{ or } C_{\text{batch}} < \theta_{C1} \\ 1 - \delta \cdot C_{\text{batch}} & \text{if } \theta_{C1} \leq C_{\text{batch}} < \theta_{C2} \\ p_{\text{min}} & \text{otherwise} \end{cases} \tag{8}$$

where $C_{\text{batch}}$ is the mean task completion rate, $V_{\text{batch}}$ is the mean action validity rate for the current batch, and $\theta$ means the transformation threshold of $p_{\text{retain}}$. The probability of retaining a positive sign, $p_{\text{retain}}$, is governed by the schedule function $f_{\text{schedule}}(C_{\text{batch}}, V_{\text{batch}})$ shown in Eqn. (8). The

**primary objective** of $f_{\text{schedule}}(C_{\text{batch}}, V_{\text{batch}})$ is to guide the sample ratio towards a healthier state as training progresses. *The hyperparameters within are set to achieve this functional goal; any function that successfully balances the sample distribution in later training stages would be suitable.* This dynamic sample balancing naturally gives rise to an **implicit two-stage curriculum**:

- **Stage 1: Learning Action Validity.** In early training, when agent performance is low, $p_{\text{retain}}$ is high. This allows the model to learn fundamental, locally valid actions from a sufficient pool of positive examples.
- **Stage 2: Learning Strategic Composition.** As the agent's proficiency increases, $p_{\text{retain}}$ decreases. This forces the agent to focus on learning the correct sequencing and strategic use of those valid actions to achieve global success.

## 4 EXPERIMENT

### 4.1 ENVIRONMENTS

To comprehensively evaluate the effectiveness of our MGR, we conduct experiments on two challenging benchmarks designed for multi-turn LLM agents: ALFWorld and AppWorld.

**ALFWorld** (Shridhar et al., 2020) is a benchmark that requires agents to perform complex, long-horizon tasks in a simulated text-based household environment. We report performance on three difficulty splits: L0, L1, and L2. Other settings are detailed in App. A.5.

**AppWorld** (Trivedi et al., 2024) is a recently proposed benchmark that evaluates agents on their ability to perform realistic, day-to-day digital tasks by interacting with a suite of simulated applications. We evaluate on its three official difficulty settings: Easy, Medium, and Difficult. Other settings are detailed in App. A.5.

Table 1: Performance comparison across ALFWorld and AppWorld benchmarks. T/A means the reward signal comes from task completion and the action correction signal.

| Model | Method | Reward Level | Reward Signal | ALFWorld | | | AppWorld | | |
|---|---|---|---|---|---|---|---|---|---|
| | | | | L0 | L1 | L2 | Easy | Medi | Diff |
| GPT-4o | - | - | - | 57.3 | 66.0 | 68.8 | 73.7 | 32.1 | 20.1 |
| DS-R1 | - | - | - | 68.8 | 70.2 | 67.3 | 85.6 | 51.2 | 29.3 |
| Qwen2.5-7B | ReAct | - | - | 23.1 | 28.5 | 27.0 | 20.2 | 6.1 | 0.0 |
| | SFT | - | - | 63.3 | 57.0 | 37.5 | 28.7 | 9.1 | 0.0 |
| | GRPO | traj | Task | 79.3 | 77.3 | 52.3 | 54.1 | 10.1 | 6.2 |
| | GiGPO | action | Task | 89.5 | **90.2** | 67.2 | - | - | - |
| | Loop | token | Task | - | - | - | 59.6 | 14.3 | 6.1 |
| | MGR | action | T/A | **90.4** | 89.7 | **85.6** | **76.1** | **19.6** | **9.4** |
| Qwen3-8B | ReAct | - | - | 28.1 | 32.5 | 29.2 | 35.1 | 8.4 | 0.0 |
| | SFT | - | - | 62.6 | 56.2 | 39.6 | 36.7 | 11.8 | 0.0 |
| | GRPO | traj | Task | 80.2 | 72.6 | 60.7 | 45.6 | 12.1 | 6.6 |
| | GiGPO | action | Task | 88.7 | **91.2** | 68.7 | - | - | - |
| | Loop | token | Task | - | - | - | 63.2 | 18.3 | 8.7 |
| | MGR | action | T/A | **91.6** | 90.8 | **82.6** | **75.4** | **22.8** | **10.2** |

### 4.2 MODELS AND BASELINES

All finetuning experiments are conducted on two powerful open-source models, **Qwen2.5-7B** (Qwen et al., 2025) and **Qwen3-8B** (Yang et al., 2025), to serve as the agent's backbone. To situate our results within the broader landscape, we also include performance metrics from the proprietary **GPT-4o** model and a previously reported strong system, **DS-R1** (DeepSeek-AI et al., 2025), as reference points for the SOTA. Our core comparisons are against a suite of finetuning baselines applied to the same models. We compare against several methods applied to the same base models for a fair and direct comparison: ReAct(Yao et al., 2022), SFT, GRPO (Shao et al., 2024), GiGPO (Feng et al., 2025), Loop (Chen et al., 2025a), detailed in App. A.3.

### 4.3 IMPLEMENTATION DETAILS

We employ the AdamW optimizer with a constant learning rate of `5e-7`. The specific hyperparameters for our Multiplicative Gated Rewards (MGR) framework are set as described in the Method section. We set the heuristic adjustment: $\beta = 0.1$ for temporal dynamics, $\alpha = 0.5$ with a threshold $q = 2$ for the repetition penalty. For the schedule function $f_{\text{schedule}}(C_{\text{batch}}, V_{\text{batch}})$ shown in Eqn. (8), we set the hyperparameters to maintain a 1:1 ratio of positive to negative samples, detailed in App. A.5. Other settings are also provided in App. A.5.

### 4.4 RESULTS AND ANALYSIS

As presented in Table 1, our proposed method consistently achieves SOTA performance across all evaluated settings on both the ALFWorld and AppWorld benchmarks. This demonstrates the effectiveness and robustness of our MGR reward framework in training capable LLM agents for complex, multi-step interaction tasks.

On the ALFWorld benchmark, while some strong baselines perform well on simpler tasks, their effectiveness noticeably drops on the more challenging L2 split. This is because L2 tasks demand longer interaction trajectories, where the correctness of each action is critical. In contrast, MGR remains robust across all difficulty levels and improves substantially on L2, indicating that early reinforcement of syntactically valid, executable actions fosters deeper and more generalizable competence.

These advantages are decisive on AppWorld. Fine-grained credit-assignment methods such as GiGPO depend on structured per-turn state descriptions to compute granular advantages. AppWorld's complex, interactive interface does not expose such a well-parsed state, making these methods **directly inapplicable** and highlighting a broader challenge in agentic RL: reward designs frequently lack generality across realistic environments. MGR, by contrast, is broadly applicable. Its action-level validity signal ($R_{\text{local}}$) is derived from generic, universally available feedback (e.g., execution errors or None responses) and lightweight syntactic checks, requiring no environment-specific state parsing. As a result, MGR can be applied seamlessly in unstructured environments like AppWorld, where it sets a new performance bar and significantly outperforms all baselines.

In summary, our empirical results strongly validate our central hypothesis: by strategically decoupling the reward signal into local validity and global effectiveness, MGR not only achieves superior and more generalizable performance but also offers a robust and universally applicable framework.

### 4.5 ANALYSIS OF TRAINING DYNAMICS

To understand MGR's effectiveness, we analyze training dynamics by comparing MGR, GRPO, and Loop baselines on the AppWorld benchmark, tracking both action success rate and task success rate. Figure 3 illustrates these learning curves, highlighting MGR's superior efficiency and stability.

The top plot of Figure 3 depicts the task success rate. MGR (red curve) exhibits a stable and consistent upward trend, achieving a much higher final success rate. This stability stems from the agent's early mastery of valid actions, providing a robust foundation for strategic learning. Conversely, Loop (blue) and GRPO (purple) show high volatility and inconsistent progress throughout training. This instability is a direct consequence of credit misassignment, where the conflated reward signals lead to contradictory policy updates. MGR's design effectively mitigates this, resulting in more reliable and efficient learning of task-level objectives. The bottom plot of Figure 3 shows the action success rate. MGR (orange curve) demonstrates rapid and significant improvement, quickly reaching over 80% validity. This reflects the efficacy of our explicit action-level validity signal ($R_{\text{local}}$), which enables efficient learning of correct action syntax. In contrast, Loop (green) and GRPO (brown) show minimal improvement, largely stagnating around 40%. Their reliance on a noisy, trajectory-level reward makes it difficult to learn basic action correctness.

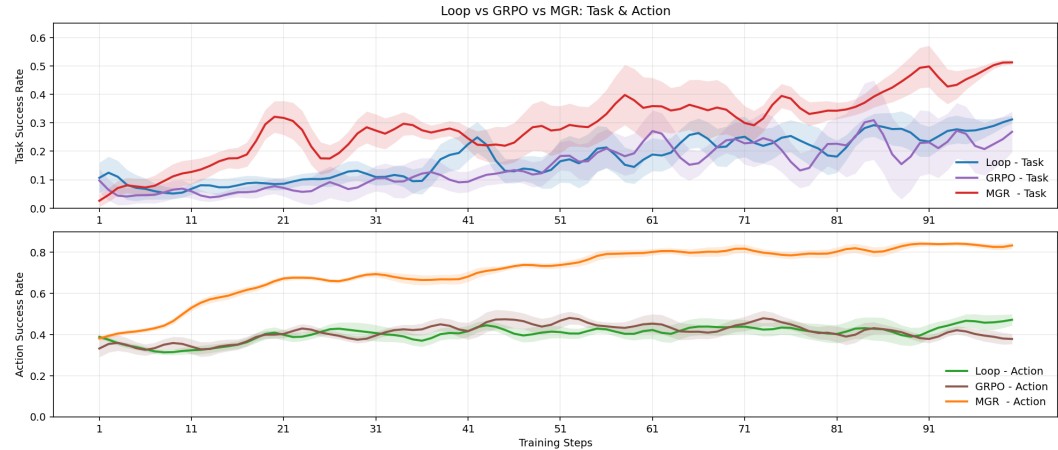

Figure 3: Training dynamics on AppWorld. Top plot: Task Success Rate. Bottom plot: Action Success Rate. MGR (red/orange) consistently outperforms Loop (blue/green) and GRPO (purple/brown) in both metrics, showing faster learning and greater stability.

## 4.6 Ablation Studies

To dissect the contribution of each key component within our MGR framework, we conduct a series of ablation studies on the AppWorld using the Qwen3-8B model. The results are presented in Table 2.

Table 2: Ablation study of MGR on the AppWorld benchmark. Each row represents the removal of a specific component from our full model.

| Method | Easy | Medium | Difficult | Avg. |
|---|---|---|---|---|
| **MGR (Full Model)** | **75.4** | **22.8** | **10.2** | **36.1** |
| - w/o Gating Mechanism ($g_t$) | 60.1 | 10.3 | 5.2 | 25.2 |
| - w/o Critic (GRPO-style Update) | 71.1 | 19.6 | 8.9 | 33.2 |
| - w/o Repetition Penalty | 15.2 | 0.0 | 0.0 | 5.1 |
| - w/ KL Divergence | 72.6 | 19.3 | 10.1 | 34.0 |

**Effect of the Gating Mechanism.** In the first ablation, we remove the stochastic gating operator $g_t$ from our reward fusion logic (Eqn. 7). This variant, labeled "- w/o Gating Mechanism", treats all valid actions in failed trajectories as positive learning samples ($g_t$ is fixed to $+1$). As observed in Table 2, this leads to a significant performance drop. This result validates our hypothesis from Section 1: without the gate, the model is overwhelmed by a massive imbalance of positive rewards for actions that are locally correct but strategically poor. The gating mechanism, which forms the basis of our implicit curriculum, is crucial for compelling the agent to transition from learning syntax to learning strategy.

**Effect of the Critic Model.** Next, we assess the role of the critic and its token-level advantage estimation in PPO. We replace our update with a GRPO-style rule in which the advantage for every token in an action is set to the final action-level reward $R_{final,t}$ (i.e., no critic model). The resulting performance degradation underscores the value of the critic in reducing the variance of policy gradients. The critic provides a more nuanced, token-level signal that stabilizes training, a benefit that is especially pronounced in the complex, high-variance environment of language-based agents.

**Effect of the Repetition Penalty.** Removing the repetition penalty results in a catastrophic performance collapse. Without the penalty, the agent quickly learns to exploit a flaw in its local optimization objective. It discovers that it can reliably accumulate positive rewards by repeatedly issuing a single, known-valid action, regardless of its strategic relevance, which lets the agent become a

repeating machine. This failure becomes absolute on Medium and Difficult tasks, as they require longer and more diverse interaction sequences.

**Effect of KL Divergence.** Our method, following prior work (Chen et al., 2025a), omits the KL divergence term commonly found in the PPO objective function. To verify this design choice, we reintroduce it in the "- w/ KL Divergence" variant. The results show a noticeable decline in performance. This suggests that the KL penalty, which constrains the policy from moving too far from the reference policy, can be overly restrictive for agentic finetuning. The complex exploration required in these tasks benefits from the greater freedom afforded by removing this constraint, allowing the policy to more aggressively adapt based on MGR's high-quality reward signals.

**Ablation on Hyperparameter Sensitivity**

**Sensitivity on Temporal Bonus ($\beta$)** To determine the optimal value for the temporal bonus hyperparameter, $\beta$, which is designed to encourage error recovery, we conducted a targeted ablation study on AppWorld. As demonstrated in Table 3, our approach is sensitive to this value. Removing the bonus entirely ($\beta = 0$) results in the lowest performance, confirming that an explicit incentive for self-correction is beneficial. A modest bonus of $\beta = 0.1$ achieves the peak success rate of 36.2%, striking an effective balance. However, we observed that larger bonus values ($\beta \geq 0.3$) lead to a slight performance degradation.

Table 3: Ablation on the temporal bonus hyperparameter $\beta$, measured by average success rate on AppWorld.

| Value of $\beta$ | Success Rate (%) |
|:---:|:---:|
| 0.0 | 32.6 |
| **0.1** | **36.2** |
| 0.3 | 36.0 |
| 0.5 | 34.6 |

This suggests that an overly strong local incentive might create a distracting signal, preventing the agent from focusing on the more crucial global strategic objective. This study empirically validates our choice of $\beta = 0.1$ for all main experiments, as it provides a useful learning signal for local error recovery while preserving the integrity of the global reward.

More experiment can be seen in App. A.6

## 5 CONCLUSION

We introduced Multiplicative Gated Rewards (MGR), a credit assignment framework that considers local action validity and global effectiveness and fuses them multiplicatively to prevent rewarding invalid actions. A fact-based local sign gates a group-relative global magnitude, yielding an implicit curriculum that first rewards producing valid actions and then prioritizes strategic composition. MGR delivers SOTA performance and markedly improved stability and sample efficiency on ALFWorld and AppWorld.

## 6 ETHICS STATEMENT

All authors have read and adhere to the ICLR Code of Ethics. Our study does not involve human subjects, private data, or potentially harmful applications; we foresee no ethical concerns beyond standard academic practices.

## 7 REPRODUCIBILITY STATEMENT

We provide anonymized source code, complete proofs, and detailed data-processing descriptions in the supplementary material to ensure full reproducibility. Hyper-parameters are listed in App. A.5.

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

# A   APPENDIX

## A.1   LLM USAGE DECLARATION

Authors prepared complete drafts of the manuscript independently. The large language model was employed solely for language polishing and stylistic refinement; no content generation, data analysis, or interpretation was performed by the LLM.

## A.2   MGR CALCULATION IN A FAILURE TRAJECTORY

To provide a comprehensive understanding of the MGR framework, we present an illustrative example of its reward calculation process within a failed trajectory, as depicted in Figure 4. This scenario serves as a direct contrast to the successful trajectory shown in Figure 2 of the main paper, highlighting MGR's core mechanism for preventing credit misassignment for actions that are locally valid but strategically ineffective.

This example demonstrates how MGR's multiplicative gating and implicit curriculum work in synergy. The mechanism ensures that in early stages, the agent is not discouraged from exploring and learning correct syntax. However, as it matures, the learning objective implicitly shifts, forcing the agent to move beyond mere syntactic correctness and master the strategic composition of actions required for long-horizon success. This prevents the model from reinforcing "echo traps" of valid but useless actions.

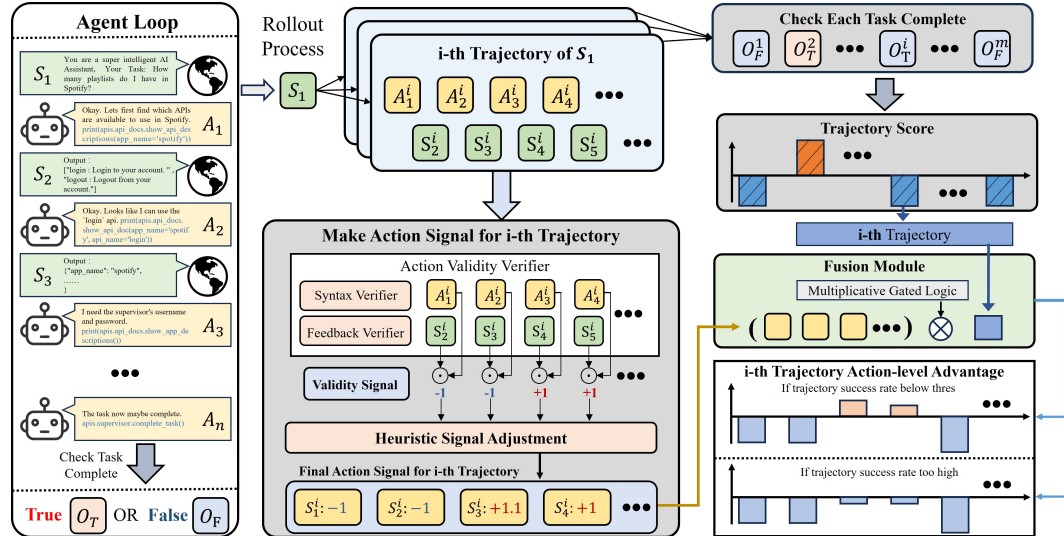

Figure 4: **Example of MGR Reward Calculation in a Failure Trajectory.** (**Top row of final rewards**) In early training, $g_t$ is likely $+1$, preserving a small positive reward to encourage learning syntax. (**Bottom row**) In late training, $g_t$ is likely $-1$, assigning a negative reward to penalize strategically poor actions and enforce learning of the correct task flow.

### A.3 BASELINES

We select the following baselines:

- **ReAct**: A zero-shot prompting baseline using the popular Reason-Act framework (Yao et al., 2022), demonstrating the base model's intrinsic agent capabilities without any parameter updates.
- **SFT (Supervised Fine-Tuning)**: Standard behavioral cloning where the model is finetuned on expert trajectories.
- **GRPO (Shao et al., 2024)**: The standard Group-Relative Policy Optimization algorithm, which applies a uniform positive or negative reward to all actions within a trajectory. This serves as our primary *trajectory-level* RL baseline.
- **GiGPO (Feng et al., 2025)**: A SOTA RL method that improves upon GRPO by incorporating an *additive turn-level* advantage estimation. This is our primary competitor representing the prevailing paradigm of credit assignment.
- **Loop (Chen et al., 2025a)**: A specialized agent method reported for the AppWorld benchmark, included as a strong contemporary baseline for that environment.

### A.4 ALGORITHM PSEUDOCODE

To formalize the training procedure, we present the MGR algorithm in pseudocode (Algorithm 1). This algorithm details the end-to-end process, from trajectory collection to the final policy update, emphasizing the crucial steps of our framework.

The key aspects highlighted in the pseudocode are:

- **Trajectory Segmentation (Lines 10–12):** Before reward calculation, each trajectory $\tau_i$ is segmented into a set of prompt-action pairs $(p'_t, a_t)$, where the prompt $p'_t$ comprises the entire history up to that point. This step correctly formats the data for training an autoregressive language model policy.
- **Decoupled Reward Calculation (Lines 14–15 & 19–22):** The algorithm computes rewards in a decoupled manner. The global, trajectory-level magnitude $R_{\text{global}}$ is computed

first (Line 15). Then, for each segmented sample, the local, action-level sign $R_{\text{local},t}$ is derived from the environmental feedback contained in the subsequent state $s_{t+1}$ (Line 20).

- **Conditional Multiplicative Fusion (Lines 24–29):** The core multiplicative gating logic is shown. The final reward $R_{\text{final},t}$ is determined by the alignment of the local and global signals, with the stochastic gate $g_t$ being the key component for the implicit curriculum.

- **Dynamic Gating (Lines 17–18 & 28):** The probability $p_{\text{retain}}$ for the stochastic gate is dynamically calculated based on the agent's batch-level performance ($C_{\text{batch}}, V_{\text{batch}}$), ensuring the reward strategy adapts as the agent improves.

---

**Algorithm 1** Multiplicative Gated Rewards (MGR) Training Loop

---

**Require:** Initial policy $\pi_\theta$, Environment $Env$
**Require:** Hyperparameters: batch size $m$, learning rate $\eta$, scaling coeff. $\gamma$
**Require:** MGR Hyperparameters: $\beta, \alpha, q$ for heuristics; $\theta_v, \theta_{C1}, \theta_{C2}, \alpha_{\text{decay}}, p_{\min}$ for curriculum
1: Initialize policy parameters $\theta$
2: **for** each training iteration **do**
3:     Initialize a raw trajectory buffer $\mathcal{B}_{\text{raw}} \leftarrow \emptyset$
4:     # — Phase 1: Trajectory Collection —
5:     **for** $i = 1$ to $m$ **do**
6:         Collect trajectory $\tau_i = (s_1, a_1, s_2, \ldots, s_{T_i}, a_{T_i})$ by executing $\pi_\theta$ in $Env$
7:         Store $(\tau_i, S(\tau_i))$ in $\mathcal{B}_{\text{raw}}$, where $S(\tau_i) \in \{0, 1\}$ is the task success signal

8:     # — Phase 2: Segmentation and Reward Calculation —
9:     Initialize a training batch buffer $\mathcal{D}_{\text{batch}} \leftarrow \emptyset$
10:     # Step 2a: Segment trajectories into (prompt, action) samples
11:     Compute $C_{\text{batch}}$ (task completion rate) and $V_{\text{batch}}$ (action validity rate) from $\mathcal{B}_{\text{raw}}$
12:     Compute positive sign retention probability $p_{\text{retain}} \leftarrow f_{\text{schedule}}(C_{\text{batch}}, V_{\text{batch}})$   ▷ Equation 7
13:     **for** each trajectory $\tau_i \in \mathcal{B}_{\text{raw}}$ **do**
14:         Compute trajectory-level magnitude $R_{\text{global}}(\tau_i) \leftarrow \frac{m}{m-1}\left(S(\tau_i) - \frac{1}{m}\sum_{j=1}^{m} S(\tau_j)\right)$   ▷ Equation 5
15:         **for** $t = 1$ to $T_i$ **do**
16:             Define prompt $p'_t \leftarrow (s_1, a_1, \ldots, s_t)$ and action $a_t$ from $\tau_i$
17:             Get next state $s_{t+1}$ from $\tau_i$ (or final state if $t = T_i$)
18:             # Step 2b: Compute action-level signals and fuse for final reward
19:             $v_t \leftarrow \text{ComputeFundamentalValidity}(a_t, s_{t+1})$   ▷ Feedback is in $s_{t+1}$
20:             $h_t \leftarrow \text{ComputeHeuristicAdjustments}(a_{1..t}, v_{1..t}, \alpha, \beta, q)$   ▷ Temporal & Repetition
21:             $R_{\text{local},t} \leftarrow v_t + h_t$   ▷ Equation 4

22:             # Multiplicative Gating Fusion Logic
23:             **if** $\text{sgn}(R_{\text{local},t}) = \text{sgn}(R_{\text{global}}(\tau_i))$ **then**
24:                 $R_{\text{final},t} \leftarrow R_{\text{local},t} \cdot |R_{\text{global}}(\tau_i)|$
25:             **else if** $R_{\text{global}}(\tau_i) > 0$ and $R_{\text{local},t} < 0$ **then**   ▷ Successful traj, invalid action
26:                 $R_{\text{final},t} \leftarrow \gamma \cdot R_{\text{local},t} \cdot |R_{\text{global}}(\tau_i)|$   ▷ Dampened penalty
27:             **else**   ▷ Failed traj, valid action
28:                 $g_t \sim \text{GatedSign}(p_{\text{retain}})$   ▷ Sample gate: $g_t \in \{+1, -1\}$
29:                 $R_{\text{final},t} \leftarrow g_t \cdot \gamma \cdot R_{\text{local},t} \cdot |R_{\text{global}}(\tau_i)|$
30:             Add $(p'_t, a_t, R_{\text{final},t})$ to training batch $\mathcal{D}_{\text{batch}}$

31:     # — Phase 3: Policy Update —
32:     Compute advantages $\hat{A}_t$ for all samples in $\mathcal{D}_{\text{batch}}$ using GAE on rewards $R_{\text{final},t}$
33:     Update policy parameters $\theta \leftarrow \theta - \eta \nabla_\theta \mathcal{L}^{\text{PPO}}(\theta)$ using samples from $\mathcal{D}_{\text{batch}}$   ▷ Equation 2

---

## A.5   Implementation Details

**Environment setting**   We utilize the ALFWorld benchmark (Shridhar et al., 2020), a text-based environment for long-horizon, embodied tasks. It is based on the ALFRED dataset and tests an

agent's ability to reason, plan, and ground language in a sequence of actions to achieve high-level goals (e.g., "put a warm apple in the microwave"). We report performance on three difficulty splits—L0, L1, and L2:

- **L0 :** Pick and Look.
- **L1 :** Clean and Heat.
- **L2 :** Cool and Pick2.

For ALFworld, we follow the setting of GiGPO (Feng et al., 2025), we use a group size of 8 and sample 16 different groups per rollout, resulting in a total of $16 \times 8 = 128$ environments. The rollout temperature is set to 1.0, the maximum response length is 512 tokens. Each episode allows up to 50 environment steps.

For AppWorld, we follow the setting of Loop (Chen et al., 2025a). We train on a subset of the AppWorld train set, excluding difficulty 3 tasks. This subset consists of 24 scenarios, with 3 minor variations (tasks) per scenario, which contain 72 tasks. Each iteration starts with the generation of K = 6 rollouts with temperature 1.0 for 40 randomly sampled tasks, for a total of 240 rollouts. We use only difficulty-1 and difficulty-2 tasks for training. The maximum response length is 2048 tokens. Each episode allows up to 35 environment steps. The task IDs we use are as follows:

```
07b42fd    229360a    27e1026    287e338    692c77d    82e2fac
aa8502b    b7a9ee9    c901732    ccb4494    ce359b5    e7a10f8
e85d92a    e3d6c94    d0b1f43    2a163ab    60d0b5b    6ea6792
29caf6f    cf6abd2    771d8fc    7d7fbf6    76f2c72    302c169
```

**Action Validity Signal ($R_{local}$) Determination**   The core of our Multiplicative Gated Rewards (MGR) framework is the ability to decouple local action validity from global trajectory success. This is primarily achieved through the action-level signal, $R_{local}$, which begins with a **Fundamental Validity Signal** ($v_t \in \{-1, +1\}$). This signal provides an immediate, fact-based judgment on the correctness of each action $a_t$. An action is considered invalid ($v_t$ is set to -1) if it fails any of the checks detailed below for each environment. Otherwise, it is considered valid ($v_t$ is set to +1).

**AppWorld**   In the AppWorld benchmark, agent actions consist of Python code designed to interact with a suite of simulated applications. An action is determined to be **invalid** if it meets one or more of the following criteria, categorized into environment feedback analysis and action format checks.

1. **Environment Feedback Analysis:** A negative validity signal ($v_t = -1$) is assigned if the environment's execution feedback contains any of the following error-indicating substrings. These keywords typically signify crashes, syntax errors, or execution timeouts:
    - `"Execution failed"`
    - `"Traceback:"`
    - `"SyntaxError"`
    - `"Exception"`
    - `"Error:"`
    - `"Maximum number of executions"`
    - `"timed out after"`
    - `"No code available to execute"`

2. **Action Format and Syntax Analysis:** Additionally, the raw action generated by the agent is preemptively checked for structural integrity. The action is considered invalid ($v_t = -1$) if it fails any of these checks:
    - **Invalid Formatting:** The generated code is not correctly encapsulated within a ```` ```python ... ``` ```` markdown block, making it unexecutable by the environment parser.

- **Absence of API Call:** The executable code, once extracted, does not contain the `.api` substring. This rule is crucial to prevent the agent from receiving rewards for syntactically correct but strategically useless code (e.g., `print("thinking")`) that does not contribute to task progression.
- **Excessive Reasoning:** The content within the `<think>` but without `</think>` tags is determined to be excessively long. This heuristic helps penalize instances where the model may be "stuck" or hallucinating, rather than producing a concise, relevant reasoning step.

**ALFWorld**  In the text-based ALFWorld environment, agent actions are natural language commands. The validity of these commands is assessed based on the environment's textual response and the agent's output format.

1. **Environment Feedback Analysis:** An action is deemed invalid ($v_t = -1$) if the textual feedback from the ALFWorld environment matches any of the following regular expression patterns. These patterns capture a wide range of common failure scenarios, such as interacting with non-existent objects or attempting illogical actions:

   - `r"^nothing happens\.?$"` (The most common feedback for an ineffectual action)
   - `r"you don't see that"` (Object is not present)
   - `r"you can't see that"` (Object is out of sight)
   - `r"that command is not understood"` (Command is not recognized by the game parser)
   - `r"you haven't got"` (Agent is not holding the required item)
   - `r"you are not"` (Agent is in the wrong location)
   - `r"you need to"` (A prerequisite condition is not met)
   - `r"you must"` / `r"you have to"` (A required condition is not fulfilled)
   - `r"that's not"` / `r"not a valid"` / `r"not valid"` (Command is syntactically or logically invalid)
   - `r"you cannot"` / `r"you can not"` (The attempted action is impossible)
   - `r"not available"` (The target object or feature cannot be used)

2. **Action Format Analysis:** A strict format is enforced on the agent's output to ensure clear separation between reasoning and action. The generated output is marked as invalid ($v_t = -1$) if it does not precisely follow the required structure, which mandates that the reasoning and the action are respectively and completely enclosed within `<think>...</think>` and `<action>...</action>` tags.

**Implicit Curriculum Hyperparameters**  For the schedule function $f_{\text{schedule}}(C_{\text{batch}}, V_{\text{batch}})$ shown in Equation 8, the hyperparameters are set to $\theta_V = 0.4$, $\theta_{C1} = 0.1$, $\theta_{C2} = 0.6$, the decay factor $\alpha = 1.5$, and the minimum retention probability $p_{\min} = 0.1$. This specific configuration is designed to counteract the inherent data imbalance in agentic learning, guiding the training process towards a healthier, near 1:1 ratio of positive to negative signals for strategically ambiguous actions.

## A.6    HYPERPARAMETER SENSITIVITY ANALYSIS

**Sensitivity on Scaling Coefficient ($\gamma$)**  We further investigated the robustness of the scaling coefficient, $\gamma$, which is used to dampen the penalty for invalid actions within successful trajectories (Scenario 2). As shown in Table 4, our method exhibits high robustness to variations in this parameter. Varying $\gamma$ within the range of $[0.8, 1.4]$ results in less than a 1% fluctuation in the final success rate compared to our default setting of $\gamma = 1.0$. This stability indicates that the precise magnitude of the penalty is less critical than the structural logic of the sign alignment. As long as the mechanism correctly identifies and penalizes

Table 4: Sensitivity analysis of the scaling coefficient $\gamma$, measured by average success rate on AppWorld.

| Value of $\gamma$ | Success Rate (%) |
|---|---|
| 0.8 | 35.4 |
| **1.0** | **36.1** |
| 1.2 | 35.9 |
| 1.4 | 35.2 |

invalid actions (preserving the negative sign), the agent effectively learns to avoid them without being overly sensitive to the specific scale of the penalty.

**Ablation on Repetition Penalty ($\alpha$)**  To assess the necessity of explicit constraints against reward hacking, we evaluated the impact of the repetition penalty coefficient $\alpha$. Table 5 reveals a critical insight: removing the penalty entirely ($\alpha = 0$) leads to a catastrophic collapse in performance (5.1%), as the agent degenerates into repeating a single locally valid action to exploit the reward signal. However, once the penalty is enabled ($\alpha \geq 0.5$), the performance stabilizes at a high level. While larger penalties (e.g., $\alpha = 2.0$) cause a minor drop by discouraging legitimate retries, the method remains effective across a wide range of non-zero values. This confirms that the presence of the penalty mechanism is a fundamental requirement for stability, but the framework is not brittle regarding its specific hyperparameter tuning.

Table 5: Impact of the repetition penalty hyperparameter $\alpha$ on App-World success rates.

| Value of $\alpha$ | Success Rate (%) |
| --- | --- |
| 0.0 | 5.1 |
| **0.5** | **36.1** |
| 1.0 | 34.2 |
| 2.0 | 34.5 |

