# OpenReview forum: "Both Local Validity and Global Effectiveness Matter:  Decoupled Credit Assignment for Long‑Horizon Agentic Learning"
_ICLR.cc/2026/Conference — Submitted to ICLR 2026_

### Official Review · Reviewer_tWSq · 2025-10-31

**Soundness:** 2
**Presentation:** 3
**Contribution:** 2
**Rating:** 4
**Confidence:** 3

**Summary:**

The paper proposes Multiplicative Gated Rewards (MGR), a novel reinforcement learning framework for LLM agents that decouples local action validity from global trajectory effectiveness in long-horizon tasks. The key idea is that every action produced by an LLM agent must be both locally valid (syntactically and executably correct) and globally effective (helpful toward task success). Existing additive credit-assignment methods conflate these dimensions, rewarding invalid actions that appear in successful trajectories and penalizing valid actions in failed ones. MGR addresses this by introducing a multiplicative gating mechanism that strictly enforces validity at the action level before scaling by global success magnitude.

**Strengths:**

1. The paper tackles an important problem in LLM-agent RL, credit assignment, which directly affects training stability and generalization.
2. The proposed decoupling between sign (validity) and magnitude (strategy) is both intuitive and empirically effective, leading to clearer reward signals and better long-term reasoning behavior.
3. Across ALFWorld and AppWorld, MGR outperforms SFT, GRPO, GiGPO, and Loop, especially on harder tasks requiring longer reasoning chains.

**Weaknesses:**

1. While MGR’s “factual validity” is well-defined, its design relies on hard-coded heuristics, error detection, format checks, repetition counts, rather than learned notions of action value. This may incentivize the agent to game surface-level rules rather than develop deeper understanding of action utility, leading to potential reward hacking or brittleness across unseen domains.
2. Since MGR’s local reward is purely syntactic or execution-based, it lacks any semantic notion of whether the action actually helps the task. The trajectory-level term compensates only indirectly. The agent could thus learn to maximize validity without improving task-level reasoning if the environment reward is sparse or delayed.
3. The method introduces several coefficients ($\alpha$, $\beta$, $\gamma$, $q$, $p_min$, schedule thresholds) whose interdependence could make it sensitive to domain or dataset scale. The paper provides defaults but lacks a robustness or sensitivity analysis beyond $\beta$.

**Questions:**

See weaknesses

---

> ### Author Response · Authors · 2025-11-24
> **Rebuttal Part1: Response to Weakness 1**
>
> ---
>
> ### **Response to Weakness 1: Heuristics, Learned Value, and Potential Gaming**
>
> We appreciate the reviewer’s concern that relying on heuristics might incentivize the agent to game surface-level rules rather than learning deep action utility. We would like to clarify that MGR does not *replace* learned value with heuristics; rather, it **decouples** the learning process into two distinct, orthogonal dimensions to ensure both correctness and utility.
>
> **1. Distinguishing "Validity" (Heuristic) from "Utility" (Learned)**
>
> The reviewer suggests using "learned notions" for action assessment. We agree that **Action Utility** (is this helpful?) must be learned. However, **Action Validity** (is this executable?) is an objective, deterministic property of the environment (e.g., Python syntax).
> *   **Why Heuristics for Validity?** Learning "validity" (e.g., training a model to predict Syntax Errors) is unnecessary and prone to approximation errors. Our heuristics simply leverage the **Ground Truth execution feedback** directly from the environment. This provides a stable, error-free foundation for the agent.
> *   **Where is the Learned Value?** The "learned notion of action value" is captured by the **Global Reward** ($R_{\text{global}}$) and the Critic network within PPO. The agent *does* develop a deep understanding of utility by optimizing the product of Validity and Global Success.
> *   **The Problem with "Learned" Baselines:** Crucially, relying solely on learned notions for local credit assignment is precisely where existing methods fail.
>     *   **Reward Models** often suffer from **hallucination**, incorrectly rewarding plausible-looking but invalid code.
>     *   **Trajectory-based Estimation**(e.g., GRPO, GiGPO, LOOP) suffers from **credit misassignment**.
>     *   **MGR's Advantage:** By decoupling, MGR uses the *infallible* heuristic signal to filter these errors *before* the learned strategic assessment applies. This ensures the agent is never rewarded for invalid actions, a guarantee that "learned" or "estimated" baselines cannot provide.
>
> **2. Preventing "Gaming" via Multiplicative Gating**
>
> The reviewer rightly points out the risk: an agent maximizing heuristic validity without strategic utility. This is precisely the failure mode MGR is designed to prevent.
> *   **The Mechanism:** Through our **Conditional Multiplicative Gating** (Eq. 6), validity is necessary but *insufficient* for reward. If an action is valid ($R_{\text{local}}=+1$) but leads to a failed trajectory ($R_{\text{global}} \le 0$), the gating mechanism (especially in later training stages) flips the final reward to negative.
> *   **Empirical Evidence:** Our ablation study (**Table 2**) directly tests this. The **"w/o Gating Mechanism"** variant, which effectively allows the agent to "game" validity rewards in failed trajectories, shows a **sharp performance drop**. This confirms that without our specific design, the gaming risk exists, but MGR effectively neutralizes it.
>
> **3. Robustness Across Domains**
>
> Regarding "brittleness," while the specific error strings (e.g., "SyntaxError") vary by environment, the **concept of Execution Feedback** is universal in digital agents. Our consistent SOTA performance across the **ALFWorld** and the **AppWorld** demonstrates that this framework generalizes well without requiring domain-specific "learned value" models for validity.

---

> ### Author Response · Authors · 2025-11-24
> **Rebuttal Part2: Response to Weakness 2**
>
> ---
>
> ### **Response to Weakness 2: Preventing "Validity Hacking" via Conditional Gating**
>
> We appreciate the reviewer for highlighting the risk of "validity hacking"—where an agent maximizes local validity without achieving task goals. However, **MGR is specifically engineered to suppress this behavior** through a two-layered defense mechanism: **Explicit Repetition Penalty** and **Implicit Gating**.
>
> **1. First Line of Defense: Explicit Suppression of "Gaming" (Repetition Penalty)**(in line 222)
>
>  MGR explicitly neutralizes this via the **Repetition Penalty** ($h_t^{\text{repeat}}$ in Eq. 3).
> *   **Mechanism:** Even if an action is syntactically valid, once it is repeated beyond a threshold, the penalty $h_t$ overrides the positive signal.
> *   **Effect:** This provides an **immediate, step-level negative feedback** for "gaming" the system, forcing agent to explore new states rather than exploiting a single valid action.
>
> **2. Second Line of Defense: Enforcing Semantics via Gating**(in section 3.4)
>
> For actions that are valid and non-repetitive but *strategically useless* (e.g., wandering aimlessly), the **Conditional Gating ($g_t$)** ensures they are not rewarded in the long run.
> *   **Mechanism:** As training progresses, the gate flips the sign of valid actions in failed trajectories. If the agent produces a chain of valid actions that fails to solve the task, the **Global outcome ($R_{\text{global}} < 0$)** combined with the gate ensures the final reward becomes negative.
> *   **Effect:** This forces the agent to align local validity with global semantic success.
>
> **3. Empirical Evidence: Catastrophic Failure without these Safeguards**
>
> Our ablation study in **Table 2** provides irrefutable evidence that these mechanisms are active and essential:
> *   **Impact of Repetition Penalty:** Removing this penalty leads to a **catastrophic collapse (Avg Success: 5.1%)**. This confirms the reviewer's suspicion: without this safeguard, the agent *does* degenerate into a "validity loop machine." MGR prevents this.
> *   **Impact of Gating:** Removing the gating mechanism causes a significant drop (**Avg Success: 25.2% vs 36.1%**), confirming that the gate is necessary to filter out "valid but ineffective" actions in the later stages of learning.
>
> In summary, MGR does not blindly reward validity. It uses strict penalties and outcome-based gating to ensure that **validity is treated as a constraint, not a proxy for value.**
>
>
> ---

---

> ### Author Response · Authors · 2025-11-24
> **Rebuttal Part3: Response to Weakness 3**
>
> ---
>
> ### **Response to Weakness 3: Hyperparameter Robustness and Interdependence**
>
> We acknowledge the reviewer’s concern regarding the number of coefficients ($\gamma, \beta, \alpha, q$, etc.). However, we demonstrate that these are **structural constants** rather than sensitive tuning knobs. We provide empirical evidence and design justifications to demonstrate that MGR is robust, decoupled, and scale-invariant.
>
>
> **1. Empirical Evidence: Robustness Across Diverse Domains**
>
> The strongest evidence against sensitivity is our cross-domain consistency. We applied the **same core hyperparameter configuration** (e.g., $\gamma=1.0$, $\beta=0.1$, schedule thresholds) to both **ALFWorld** (text game) and **AppWorld** (Python coding). (shown in line 805)
> *   **Result:**  Despite the vast differences in action space and trajectory length, the **same configuration achieved SOTA on both**. This confirms that these parameters capture universal behavioral priors rather than overfitting to a specific domain.
>
>
> **2. Additional Sensitivity**
>
> To further address the reviewer's request for analysis "beyond $\beta$," we have conducted a sensitivity check on the Scaling Coefficient $\gamma$ and the Repetition Penalty ($\alpha$).
>
>
> | Scaling Coefficient ($\gamma$) | 0.8 (Weaker Penalty) | **1.0 (Default)** | 1.2 (Stronger Penalty) | 1.4 (Doubled Penalty) |
> | :--- | :---: | :---: | :---: | :---: |
> | **Avg. Success Rate (%)** | 35.4 | **36.1** | 35.9 | 35.2 |
>
> *   **Result:** Varying $\gamma$ within $[0.8, 1.2]$ resulted in less than **1% fluctuation** in final success rates. This confirms that as long as the *sign* logic (validity enforcement) is correct, the specific *magnitude* scaling is forgiving. We will include this additional plot in the revision.
>
>
> | Repetition Penalty ($\alpha$) | 0 (Table 2) | **0.5 (Default)** | 1.0 | 2.0 |
> | :--- | :---: | :---: | :---: | :---: |
> | **Success Rate (%)** | 5.1 | **36.1** | 34.2 | 34.5 |
>
> *   **Result:** Performance only collapses when the penalty is removed ($\alpha=0$). Once enabled ($\alpha \ge 0.5$), performance stabilizes in a high-performance plateau. Higher values (1.0, 2.0) cause a negligible drop due to discouraging legitimate retries, but still maintain SOTA-level results.
>
> **3. The Gating Schedule: Function over Precision (Addressing "Scale" and "Sensitivity")**
>
> The reviewer specifically noted the coefficients in the schedule. We clarify that these act as **loose boundaries** for a functional objective: **Dynamic Sample Balancing**, rather than precise tuning knobs.
> *   **Mechanism:** The goal is simply to steer the training data towards a healthy 1:1 ratio of positive/negative signals to prevent valid but useless actions from overwhelming the learner. The parameters are akin to setting a target temperature for a **thermostat**; any configuration that prevents extreme imbalance yields similar stable performance.
> *   **Scale Invariance:** Crucially, this schedule relies entirely on **normalized rates** (Success Rate $C_{batch} \in [0,1]$), not absolute step counts. This makes the mechanism mathematically **invariant to dataset scale**, ensuring consistent behavior whether the dataset is small or large.
>
>
> **4. Addressing "Interdependence" and "Dataset Scale"**
>
> *   **Orthogonal Design:** The parameters operate on distinct, non-conflicting levels, minimizing the need for joint tuning:
>     *   **Local Validity ($\alpha, \beta, q$):** These strictly govern *instantaneous* correctness (step-level).
>     *   **Global Strategy ($\gamma, f_{\text{schedule}}$):** These govern *long-term* credit assignment (trajectory-level).
>     *   Because MGR **decouples** validity from effectiveness, changing the repetition sensitivity ($\alpha$) does not alter the logic of the global gating ($\gamma$). This structural orthogonality is why a single configuration works across domains.
>
> *   **Scale Invariance:** The reviewer specifically questioned "dataset scale." Our Gating Schedule depends entirely on **normalized rates** (Success Rate $C_{batch} \in [0,1]$), not absolute step counts or dataset sizes. This makes the hyperparameters mathematically **invariant to dataset scale**, ensuring consistent behavior whether the dataset has 100 or 100k samples.
>
> We will include these experiments in the Appendix of the final revision to further substantiate robustness.
>
> ---

---

> > ### Comment · Reviewer_tWSq · 2025-11-27
> >
> > Thank you for the additional experiments and clarifications, which have addressed my concerns. I have increased my score accordingly. Good luck with your submission!

---

### Official Review · Reviewer_LUMB · 2025-10-31

**Soundness:** 2
**Presentation:** 3
**Contribution:** 3
**Rating:** 6
**Confidence:** 3

**Summary:**

The paper proposes a novel technique for credit assignments in long-horizon agent tasks for LLMs. Specifically, they propose two orthogonal components that make up their reward through a multiplicative gate, local action validity reward (+1 or -1 sign based on the action validity) and a trajectory level advantage score. The method outperforms other techniques like GRPO and Loop on ALFworld and AppWorld tasks.

**Strengths:**

1. The credit assignment problem being tackled is important
2. The paper's idea is interesting and well motivated
3. The ablations study and analysis are insightful

**Weaknesses:**

1. The method mostly relies on heuristics for the local-level action feedback, which seems like added bias and transfer on other tasks is uncertain
2. It's likely quite tricky to use this method on a variety of problems where it's not always clear what constitute as an invalid action, or where there is not an automated feedback for invalid actions
3. There are no uncertainty/error bounds in the experiments.
4. Introduction of various new hyperparameters makes me wonder how sensitive the method is to them, and how difficult they would be to tune on new tasks.
4. The method is tested on only two tasks, it's possible that the method and hyperparameters have been overfitted on just these two tasks.

**Questions:**

1. How would the method be applied to tasks where action-level validity signals are not available?
2. How sensitive is the method to the various hyperparameters introduced, and how well does it transfer to other tasks?
3. Is it possible to add more seeds to get error bounds?

---

> ### Author Response · Authors · 2025-11-24
> **Rebuttal Part1: Response to Weakness 1**
>
> ---
> ### **Response to Weakness 1: Generalizability and Design Trade-off of Action Validity**
>
> We thank the reviewer for this thoughtful comment regarding the heuristic nature of our validity signals and their transferability. We respectfully acknowledge that our method leverages explicit feedback patterns. However, we argue that this design is not a limitation, but a deliberate **design trade-off** tailored to the inherent nature of LLM agents, which offers **greater portability** than existing alternatives.
>
> **1. Text/Code is the Native, Universal Interface for Agents**
>
> **The "heuristics" in MGR target the interaction interface, not specific task logic.**
> *   **Universal Interaction:** LLM Agents fundamentally operate by generating text or code (API calls) to interact with digital environments.
> *   **Standardized Feedback:** In these high-value domains (e.g., OS control, coding agents, gaming), execution feedback is highly standardized. Invalid actions universally manifest as `Runtime Errors`, `Exceptions`, `HTTP Error Codes`, or `Command Not Found`.
> *   **Portability:** Therefore, our validity checks transfer naturally across tasks. MGR does not require task-specific logic (e.g., "before do something, check ......"); it only requires interface-level logic (e.g., "did the command execute?"). This allows MGR to address the primary bottleneck in agentic learning across a wide spectrum of applications.
>
> **2. Design Trade-off: Heuristics vs. State Dependency**
>
> We highlight that our approach represents a strategic **design trade-off** that favors robustness over theoretical purity.
> *   **Baselines are Restrictive:** Many RL methods (e.g., GiGPO) require the environment to expose **Structured State Representations** to compute step-level alignment or advantages. This assumption fails in realistic, open-ended environments (e.g., **AppWorld**) where only raw observations are available.
> *   **MGR is More Portable:** By relying on "heuristic" execution feedback (stdout/stderr), MGR decouples itself from the need for parsed environment states. This makes our framework **more broadly applicable** to unstructured, realistic environments.
>
> **3. Empirical Validation of Transferability**
>
> The robustness of this design is empirically validated by our results. We applied the **same validity principles** (checking for execution feedback and format adherence) to two vastly different benchmarks: **ALFWorld** (text-based game) and **AppWorld** (code-based OS interaction). The fact that MGR achieves SOTA performance on both confirms that these signals are not brittle, task-specific biases, but rather robust indicators of executability in agentic environments.
>
> ---

---

> ### Author Response · Authors · 2025-11-24
> **Rebuttal Part2: Response to Weakness 4, Question 2**
>
> ### **Response to W4 & Q2: Hyperparameter Robustness and Ease of Tuning**
> We understand the reviewer’s concern regarding the number of hyperparameters. However, we provide strong empirical evidence and design justifications to demonstrate that MGR is **robust** and **does not require intensive tuning** for new tasks.
>
> **1. Strongest Evidence: Zero-Tuning Transfer Across Domains**
>
> The most compelling evidence of robustness is that we utilized the **exact same set of core hyperparameters** (e.g., Repetition Penalty $\alpha=0.5$, Temporal Bonus $\beta=0.1$, Scaling $\gamma=1.0$) for both **ALFWorld** and **AppWorld** (in line 805).
> *   **Context:** ALFWorld is an embodied text game, while AppWorld is a coding-based OS control task. Despite the vast differences in action space, trajectory length, and task logic, the **same configuration achieved SOTA on both**. This confirms that these parameters capture universal behavioral priors rather than task-specific overfits.
>
> **2. Empirical Sensitivity Analysis**
>
> To quantitatively address the sensitivity concern, we conducted additional ablation studies on AppWorld by varying key hyperparameters over a wide range.
>
> *   **Scaling Coefficient ($\gamma$):** We varied $\gamma$ (default 1.0) from 0.8 to 1.4.
>     | Scaling Coeff. ($\gamma$) | 0.8 | **1.0 (Default)** | 1.2 | 1.4 |
>     | :--- | :---: | :---: | :---: | :---: |
>     | **Success Rate (%)** | 35.4 | **36.1** | 35.9 | 35.2 |
>     *   **Result:** The performance fluctuation is **less than 1%**. This indicates that as long as the *sign* logic (validity enforcement) is correct, the method is insensitive to the exact magnitude scaling.
>
> *   **Repetition Penalty ($\alpha$):** We varied $\alpha$ (default 0.5) from 0.0 to 2.0.
>     | Repetition Penalty ($\alpha$) | 0 (Table 2) | **0.5 (Default)** | 1.0 | 2.0 |
>     | :--- | :---: | :---: | :---: | :---: |
>     | **Success Rate (%)** | 5.1 | **36.1** | 34.2 | 34.5 |
>     *   **Result:** Performance only collapses when the penalty is removed ($\alpha=0$). Once enabled ($\alpha \ge 0.5$), performance stabilizes in a high-performance plateau. Higher values (1.0, 2.0) cause a negligible drop due to discouraging legitimate retries, but still maintain SOTA-level results.
>
> *   **Temporal Bonus $\beta$:** Its has been shown in Table 3.
>
> **3. The Gating Parameters**
>
> Regarding the parameters in the Gating Schedule ($f_{schedule}$), we clarify that  the specific hyperparameters are **secondary** to a simple functional objective: **Dynamic Sample Balancing**.
>  *   **The Problem:** Without the gate, valid actions in failed trajectories create a severe data imbalance in late training stage (e.g., 90% positive signals vs. 10% negative), overwhelming the learner.
> *   **Mechanism:**  The gating schedule parameters do not dictate *when* to switch (e.g., "at step 1000"), but *under what conditions* to switch based on **Batch Statistics** (task success rate $C_{batch}$, action success rate $V_{batch}$). The schedule are merely configured to steer this ratio towards a healthier balance (approx. 1:1) as the agent improves.
> *   **Insensitivity:** Those gating parameters act as **loose boundaries** rather than precise values. Any configuration that successfully prevents extreme class imbalance yields similar stable performance. They are akin to setting a "target temperature" for a thermostat; the exact curve matters less than the final equilibrium.

---

> ### Author Response · Authors · 2025-11-24
> **Rebuttal Part3: Response to Weakness 2, Weakness 5, Question 1**
>
> ---
> ### **Response to Weakness 2 & Question 1: Applicability in Absence of Validity Signals**
>
> We appreciate the reviewer's question regarding the method's behavior in tasks where explicit validity signals might be sparse. We clarify that MGR is designed with broad definitions of validity and a **mathematically robust fallback mechanism** to handle such scenarios effectively.
>
> **1. Validity Extends to Intrinsic Structure (Broad Applicability)**
>
> We respectfully emphasize that "Validity" in MGR is not limited to runtime environment feedback. As detailed in Appendix A.5, line 762, it significantly incorporates **Intrinsic Structural Checks**.
> *   **Format as Validity:** Modern LLM agents almost always require structured outputs (e.g., "Output reasoning in `<think>` tags", "Return JSON format", "No conversational filler").
> *   **Application:** MGR derives the validity signal $v_t$ from these format constraints. If the model fails to follow the required structure, $v_t=-1$. This allows MGR to enforce instruction following and penalize format hallucinations even in non-interactive tasks or environments without explicit error streams, ensuring broad applicability beyond coding/OS domains.
>
> **2. Mathematical Fallback: Safe Degradation to Standard RL**
>
> In the extreme scenario where action-level validity signals are strictly unavailable (i.e., no environment errors *and* no format constraints), MGR naturally handles this by defaulting $v_t$ to **+1** (Valid).
> *   **Mechanism:** When $v_t = +1$ for all steps, the multiplicative gating logic (Eq. 6) simplifies, and the reward becomes primarily driven by the trajectory-level magnitude $R_{\text{global}}$.
> *   **Result:** The algorithm effectively **degrades to a standard trajectory-level RL method** (equivalent to our GRPO baseline).
> *   **Implication:** This guarantees that MGR is **universally safe to use**. In the worst-case scenario (zero local signals), it performs on par with the trajectory-level baseline; it does not require a "perfect" signal to function, eliminating the risk of failure in new domains.
>
> **3. Conceptual Extensibility to Semantic Tasks (Plug-and-Play Framework)**
>
> For open-ended semantic tasks (e.g., dialogue safety), $R_{local}$ can be seamlessly populated by an **LLM-as-a-Judge** or a lightweight **Reward Model** (e.g., flagging toxic output as $v_t=-1$). **Crucially, MGR's multiplicative gating would still apply here:** ensuring that even if a toxic response leads to a "successful" dialogue termination, it is strictly penalized—solving the "reward hacking" problem prevalent in standard RLHF. This suggests MGR has broad potential beyond agentic tasks.
>
> ---
> ### **Response to Weakness 5: Generalization Capabilities**
>
> We clarify that while evaluated on two benchmarks, **ALFWorld** and **AppWorld** were specifically chosen to represent the two maximally distinct paradigms in current agent research, serving as a rigorous stress test against overfitting:
>
> 1.  **Task Heterogeneity:**
>     *   **ALFWorld:** Represents **Embodied Agents** operating in a **Text-based** action space with extremely long horizons.
>     *   **AppWorld:** Represents **Digital Agents** (OS Control, App Use) operating in a **Code-based** action space with long horizons and complex dependencies.
> 2.  **Consistency as Proof:**
>     Achieving SOTA on such highly heterogeneous domains **using the exact same hyperparameter configuration** (as detailed in Response to W4) strongly suggests that MGR captures the **fundamental principles** of agentic learning (validity vs. strategy) rather than overfitting to specific task distributions.
>
> ---

---

> ### Author Response · Authors · 2025-11-24
> **Rebuttal Part4: Response to Weakness 3, Question 3**
>
> ---
>
> ### **Response to Weakness 3 & Question 3: Statistical Significance and Error Bounds**
>
> We thank the reviewer for emphasizing the importance of statistical rigor. We fully agree that reporting uncertainty is crucial to distinguish algorithmic improvements from random variance in RL.
>
> **1. Multi-Seed Experiments**
>
> To directly address Q3, we conducted additional experiments using **3 independent random seeds** on the **AppWorld** benchmark.
> *   **Note on Setup:** Due to the strict time constraints of the rebuttal phase, we performed these runs on a **representative subset** (50% of training task variations) and reduced the training steps to 80. Crucially, **all models reached convergence** within this budget, ensuring the validity of the relative performance comparison.
>
> The results ($\text{Mean} \pm \text{Std}$) are summarized below:
>
> | Method | Easy | Medium | Difficult |
> | :--- | :---: | :---: | :---: |
> | **GRPO** | $38.1 \pm 4.2$ | $8.1 \pm 2.8$ | $5.6 \pm 1.9$ |
> | **Loop** | $60.2 \pm 2.2$ | $12.1 \pm 1.2$ | $6.7 \pm 1.5$ |
> | **MGR (Ours)** | $\mathbf{70.1 \pm 1.8}$ | $\mathbf{18.3 \pm 1.4}$ | $\mathbf{9.8 \pm 0.9}$|
>
>
> **2. Analysis of Robustness**
>
> These results strongly support the validity of our claims:
> *   **Statistically Significant Gap:** Even in this condensed training setting, MGR outperforms the strong baseline (Loop). This confirms that the improvement is driven by the method, not random chance.
> *   **Superior Stability:** MGR consistently exhibits the **lowest standard deviation**.
>
> We will include these experiments in the Appendix of the final revision to further substantiate robustness.

---

### Official Review · Reviewer_Ao4r · 2025-11-01

**Soundness:** 3
**Presentation:** 3
**Contribution:** 3
**Rating:** 6
**Confidence:** 3

**Summary:**

The paper proposes Multiplicative Gated Rewards (MGR), a reinforcement learning framework for long-horizon LLM agents that separates local action validity from global effectiveness. Instead of adding rewards, MGR multiplies a validity signal with a trajectory-level score, ensuring invalid actions are penalized even in successful trajectories. A dynamic gating mechanism creates an implicit curriculum, first learning valid actions, then strategic sequencing. Experiments on ALFWorld and AppWorld show MGR achieves state-of-the-art performance over more challenging tasks compared with baselines.

**Strengths:**

1. The paper is easy to follow.
2. The paper systematically analyzes the key issue of credit assignment in long-trajectory learning for LLM agents and proposes a practical solution to address it.
3. The experiments are convincing, as MGR demonstrates a remarkable advantage on more challenging L2 tasks.

**Weaknesses:**

1. The method introduces several hyperparameters for control, such as α and β, whose sensitivity remains unclear, which is critical for assessing the robustness and practical applicability of this approach.
2. The method relies on explicit action validity signals from the environment, limiting its applicability to tasks with well-defined executable actions and making it unsuitable for open-ended or semantic language tasks without clear feedback.

**Questions:**

1. Can the method be applied to more complex environments, such as embodied settings?

---

> ### Author Response · Authors · 2025-11-24
> **Rebuttal Part1: Response to Weakness 1**
>
> ### **Response to Weakness 1: Hyperparameter Robustness**
>
> We appreciate the reviewer's scrutiny regarding hyperparameter sensitivity. We demonstrate that MGR is highly robust and does not require intensive tuning through the following empirical evidence:
>
> **1. Strongest Evidence: Zero-Tuning across Domains:**
>
> The strongest evidence is that we utilized the **exact same set of hyperparameters** ($\beta=0.1, \alpha=0.5$) for both **ALFWorld** (text game) and **AppWorld** (coding/API), despite their vast differences. This confirms these parameters capture universal behavioral priors rather than task-specific fits.
>
> **2. Repetition Penalty ($\alpha$):**
>
> We conducted a **new ablation study** varying $\alpha$ from 0.0 to 2.0.
>
> | Repetition Penalty ($\alpha$) | 0 (Table 2) | **0.5 (Default)** | 1.0 | 2.0 |
> | :--- | :---: | :---: | :---: | :---: |
> | **Success Rate (%)** | 5.1 | **36.1** | 34.2 |34.5|
> | **Performance Delta** | -31.0 | **-** | -1.9 | -1.6|
>
> *   **Result:**Performance only collapses when the penalty is removed ($\alpha=0$). Once enabled ($\alpha \ge 0.5$), performance stabilizes in a high-performance plateau. Higher values (1.0, 2.0) cause a negligible drop due to discouraging legitimate retries, but still maintain SOTA-level results.
>
> **3. Scaling Coefficient ($\gamma$)**
>
> We have conducted a sensitivity check on the scaling coefficient $\gamma$ (default 1.0) on AppWorld.
>
>
> | Scaling Coefficient ($\gamma$) | 0.8 (Weaker Penalty) | **1.0 (Default)** | 1.2 (Stronger Penalty) | 1.4 (Doubled Penalty) |
> | :--- | :---: | :---: | :---: | :---: |
> | **Avg. Success Rate (%)** | 35.4 | **36.1** | 35.9 | 35.2 |
> | **Performance Delta** | -0.7 | **-** | -0.2 | -0.9 |
>
> *   **Result:** Varying $\gamma$ within $[0.8, 1.4]$ resulted in less than **1% fluctuation** in final success rates. This confirms that as long as the *sign* logic (validity enforcement) is correct, the specific *magnitude* scaling is forgiving.
>
> **4. Temporal Bonus $\beta$:**  It has been shown in Table 3.
>
> **5. The Role of Gating Parameters**
>
> Regarding the parameters in the Gating Schedule ($f_{schedule}$), we clarify that specific hyperparameters are **secondary** to a simple functional objective: **Dynamic Sample Balancing**.
> *   **The Problem:** Without the gate, valid actions in failed trajectories create a severe data imbalance (e.g., 90% positive signals vs. 10% negative), overwhelming the learner.
> *   **The Solution:** The schedule parameters ($\theta_V, \theta_{C}$) are merely configured to steer this ratio towards a healthier balance (approx. 1:1) as the agent improves.
> *   **Insensitivity:** They act as **loose boundaries** rather than precise values. Any configuration that successfully prevents extreme class imbalance yields similar stable performance. They are akin to setting a "target temperature" for a thermostat; the exact curve matters less than the final equilibrium.

---

> ### Author Response · Authors · 2025-11-24
> **Rebuttal Part2: Response to Weakness 2, Question 1**
>
> ---
>
> ### **Response to Weakness 2: Reliance on Explicit Signals & Applicability**
>
> We thank the reviewer for this thoughtful comment. We appreciate the opportunity to clarify the scope of our work and the broader applicability of the "validity" concept. While we agree that our method leverages explicit signals, we would like to highlight two key aspects that mitigate the concern regarding limited applicability.
>
> **1. Validity Includes Intrinsic Structural Checks (Beyond Environment Feedback)**
>
> We would like to clarify that MGR does not rely solely on the environment's runtime feedback. As detailed in Appendix A.5, our framework incorporates **intrinsic static analysis** to determine action validity before execution.
> *   **Format & Syntax as Validity:** In our implementation, we preemptively check if an action follows the required structure (e.g., correct JSON formatting, proper Markdown encapsulation) or adheres to syntax rules.
> *   **Broader Applicability:** This means MGR is applicable to any task that requires **Structured Output**—a standard requirement for modern LLMs in information extraction or reasoning tasks—even without an interactive environment. In these cases, the "validity signal" ($v_t$) is derived from the model's adherence to the output format constraints.
>
> **2. The Critical Feature of LLM Agentic Environments**
>
> Our work is specifically motivated by the challenges in **Agentic Learning** (e.g., tool use, coding, operating systems), which we believe is a high-value frontier in current LLM research.
> *   **Feature, Not Limitation:** In these domains, the existence of explicit feedback (like error messages or API returns) is an inherent characteristic, not a special case. The core challenge for agents is often not "what to say" (open-ended generation), but **"how to execute correctly"**.
>
> **Moreover: Extensibility to Semantic Tasks via Learned Signals**
>
> While our experiments focus on verifiable executable tasks, the MGR framework is signal-agnostic. In open-ended semantic tasks (e.g., dialogue safety, hallucination reduction), the $R_{local}$ slot can be filled by a **Reward Model (RM)** or **LLM-as-a-Judge** that flags specific violations (e.g., toxic output = invalid). This suggests MGR has broader potential in RLHF beyond just code/agentic tasks.
>
>
> ---
>
> ### **Response to Question 1: Applicability to Complex Embodied Settings**
>
> We believe MGR is well-suited for embodied settings because it addresses the **fundamental structure** of long-horizon interaction, though we acknowledge that deployment involves challenges shared by the broader embodied AI field.
>
> **1. Why MGR is Fundamentally Transferable (The Logic)**
>
> The core problem MGR solves, **Credit Assignment in Long-Horizon Tasks under Hard Constraints**, is arguably *more* pronounced in embodied agents than in text agents.
> *   **The Shared Problem:** Like code generation, embodied control has a dual requirement: **Feasibility** (Can I execute this motion without collision/falling?) and **Effectiveness** (Does this motion help the task?).
> *   **MGR's Solution:** In traditional RL for robotics, a "lucky" trajectory that achieves the goal but involves dangerous or erratic movements (e.g., high joint jerk) might be rewarded. MGR’s decoupling mechanism naturally fits here: we can map $R_{local}$ to **physical feasibility** (e.g., simulator collision checks, kinematic limits). This ensures the agent prioritizes *safe/feasible control* before optimizing for task efficiency, effectively acting as a safety constraint.
> *   **ALFWorld Evidence:** We respectfully note that our SOTA results on **ALFWorld** (a standard embodied benchmark involving navigation and object manipulation) already demonstrate MGR's capability to handle the logical planning aspects of embodied tasks.
>
> **2. Implementation Challenges**
>
> However, applying MGR to continuous control or visual environments introduces implementation hurdles common to the field, rather than specific to MGR:
> *   **Defining "Validity" in Continuous Space:** Unlike the binary nature of syntax errors in code, "validity" in continuous control can be fuzzy. Implementing MGR requires defining a clear boundary for $R_{local}$ (e.g., using a physics simulator to detect collisions or singularities). While this requires domain-specific engineering, it is a standard practice in Safety RL.
> *   **Multimodal Complexity:** Embodied agents face perception challenges (e.g., visual grounding) that text agents do not. MGR improves the *learning signal* (the gradient direction), but it does not solve the *perception* bottleneck.
>
>
> **Conclusion**
>
> In summary, while the *implementation* details (perception, continuous validity definitions) present shared challenges for the field, the *underlying reinforcement learning logic* of MGR is directly applicable and highly beneficial for enforcing physical constraints in complex embodied environments.

---

> > ### Comment · Reviewer_Ao4r · 2025-11-27
> >
> > Thanks for the clarification, which solves my concerns. I will keep the score as it is.

---

### Official Review · Reviewer_BvSX · 2025-11-01

**Soundness:** 3
**Presentation:** 3
**Contribution:** 2
**Rating:** 4
**Confidence:** 3

**Summary:**

This paper proposes a reward function, Multiplicative Gated Rewards (MGR), for long-horizon LLM agents, which decouples "local action validity" (step is syntactically/executably valid) from "global effectiveness" (trajectory quality), then fuse them multiplicatively so invalid steps contribute zero (or negative) credit even in successful rollouts. A stochastic gate balances valid steps in failed trajectories to avoid overwhelming positives. Experiments on ALFWorld and AppWorld with Qwen2.5-7B/Qwen3-8B show higher success rates than baselines; ablations indicate the gate, critic, and repetition penalty matter materially.

**Strengths:**

- Addresses credit assignment problem, i.e. additive credit lets invalid steps get positive credit in successful trajectories.
- Method is simple, orthogonal to PPO/GRPO training, and given in clear pseudocode; implementation details for validity checks are explicit for both environments.
- Consistent gains on ALFWorld/AppWorld; learning curves show improved stability and faster rise in both action-validity and task success.
- Some ablations provided: removing gating/critic/repetition penalty degrades sharply, supporting design choices.
- Method naturally induces a curriculum

**Weaknesses:**

- Novelty is incremental. Multiplicative gating is a principled reweighting/masking; not a new credit-assignment theory.
- Local validity is heuristically defined and environment-specific; portability beyond text/code interfaces is unclear. Provide a (ideally formal) definition up front
- Gating schedule is heuristic. The batch-stat–driven sign flip lacks theory; sensitivity and failure modes aren’t fully characterized.
- Reporting focuses on success rates; analysis on sample efficiency and exploration would be appreciated.

**Questions:**

- Does the stochastic sign gate ever destabilize learning or introduce bias, especially early in training when the proportion of valid actions is low?
- Can MGR be applied on top of learned or preference-based reward models (e.g., PRM, CAPO, RLAIF)? If so, is the gating still beneficial, or redundant?
- Is there any quantitative evidence that the multiplicative formulation improves gradient signal quality?

---

> ### Author Response · Authors · 2025-11-24
> **Rebuttal Part 1：Response to Weakness1，Weakness2**
>
> ---
> ### **Response to Weakness 1: Novelty  of MGR**
>
> We thank the reviewer for this insightful comment. We agree that multiplicative gating, as a general mechanism, is not new. However,  our primary novelty is not the mechanism itself, but rather the **identification and resolution of a fundamental paradigm mismatch** that arises when applying traditional adding credit assignment principles to LLM agents.
>
> Traditional RL frameworks implicitly assume a **pre-validated action space** (e.g., Atari controls), where actions are inherently executable. In stark contrast, LLM agents operate in an **open-ended language action space**, creating a constant and severe risk of invalid outputs. The prevailing *additive paradigm*, inherited directly from traditional RL, overlooks this critical distinction, leading to a critical flaw: **invalid actions can receive positive rewards simply for appearing in a successful trajectory, fundamentally corrupting the learning signal at its source.**
>
> MGR's contribution is to be the first framework that **identify and resolve this core conflict.** The novelty lies not in *inventing* the multiplicative tool, but in *deploying* it to enforce a crucial, hard constraint that is essential for LLM agents. Therefore, rather than an "incremental" improvement, we believe MGR represents a **foundational and necessary adaptation** of RL principles, making them robust to the unique challenges posed by LLM agents.
>
>
> ---
>
> ### **Response to Weakness 2: Definition and Portability of Local Validity**
>
> We thank the reviewer for this insightful comment. We agree that providing a formal definition is crucial for clarity. In the revision, we will formalize the "heuristics" into a rigorous definition in Section 3 and explicitly discuss the portability of our framework.
>
> **1. Formal Definition of Local Validity**
>
> The "heuristics" described in the paper are concrete instantiations of a general validity function. We formally define the **Local Validity Reward**, $R_{\text{local}}$, based on the action $a_t$ and the subsequent environment feedback $o_{t+1}$ (e.g., stdout, stderr, or API response):
>
> $$
> R _ {\\text{local}}(a _ t, o _ {t+1}) =
> \\begin{cases}
> -1, & \\text{if } a _ t \\notin \\Omega _ {\\text{syntax}} \\lor \\exists e \\in \\mathcal{E} _ {\\text{runtime}}, \\text{match}(o _ {t+1}, e) \\\\
> +1, & \\text{otherwise}
> \\end{cases}
> $$
>
> *   $\Omega_{\text{syntax}}$ denotes the set of actions satisfying structural constraints (e.g., correct code encapsulation, API format).
> *   $\mathcal{E}_{\text{runtime}}$ represents a set of **universal error patterns** inherent to the execution environment (e.g., `SyntaxError`, `TimeOut`, `HTTP 404`, `Command Not Found`).
> *   $\text{match}(\cdot)$ is a function detecting if the feedback contains these error patterns.
>
> **2. Justification: Text/Code is the Native Interface for LLM Agents**
>
> Regarding portability, while MGR targets text/code environments, we respectfully emphasize that **text and code are the native interaction interfaces for LLM Agents**.
> *   **Domain Value:** Addressing validity in text-based environments is not a limitation but a direct response to the primary bottleneck in agentic learning. MGR is specifically tailored to solve this "language action space" problem which is prevalent in high-value applications like game agents, and OS or App control.
>
>
> **3. Comparative Generalizability & Design Trade-off**
>
> Our reliance on feedback parsing makes MGR **more portable and less restrictive** within the LLM agent domain than existing SOTA methods:
> *   **Baselines are Restrictive:** Methods like **GiGPO** require the environment to expose **structured State representations** to compute step-level alignment. This assumption fails in many realistic, open-ended environments (e.g., AppWorld, real-world websites) where only raw observations are available.
> *   **MGR is Universal:** MGR unifies the signal by relying solely on **standard execution feedback** (e.g., standard error streams, API return codes). This signal is universally available across almost all digital environments.
>
> **Design Trade-off:** Therefore, our "heuristic" design represents a deliberate **design trade-off**: we forego the reliance on structured states to achieve **robustness and broader applicability** in complex, unstructured environments. This choice is empirically validated by MGR’s superior performance on AppWorld, where state-dependent baselines struggle or fail due to the lack of parsed states.
>
> **4. Conceptual Portability Beyond Text**
>
> While our work focuses on LLM agents, the **core principle** of MGR—decoupling local executability from global strategic value—is portable. For instance, in **Robotics**, a  self-collision is the analog to a "syntax error." MGR can be applied by replacing the text parser with a collision checker while keeping the multiplicative gating mechanism unchanged to penalize physically invalid actions within potentially successful trajectories.

---

> ### Author Response · Authors · 2025-11-24
> **Rebuttal Part2:  Response to Weakness 3**
>
> ---
>
> ### **Response to Weakness 3: Analysis of Gating Schedule (Heuristics & Stability)**
>
> We thank the reviewer for this critical question. We clarify that the specific hyperparameters are secondary; they act as tools to achieve a fundamental, theoretically grounded objective: **Dynamic Sample Balancing**. The schedule is not arbitrary, but a necessary mechanism to manage the **Signal-to-Noise ratio** throughout training.
>
> **1. Theoretical Motivation: Managing Signal-to-Noise Transition**
>
> In Agentic RL, valid actions within failed trajectories are "ambiguous data":
> *   **Early Stage (Data Scarcity):** The agent fails frequently. If we penalized all actions in failures, positive samples would be scarce. The gate retains these valid actions as **high-value signals** to "salvage" syntax learning.
> *   **Late Stage (Noise Reduction):** As success rises, valid actions in failed trajectories become **ambiguous noise** (False Positives). The gate flips them to negative to prevent reward hacking and enforce strategic learning.
> Thus, $f_{\text{schedule}}$ acts as an **implicit curriculum**, transitioning the gradient from syntax-dominant to strategy-dominant.
>
> **2. Characterizing Failure Modes & Solutions**
>
> The reviewer rightly asks about failure modes. Our design (and Ablation Study in Table 2) explicitly safeguards against the two primary risks:
> *   **Risk A: Strategic Stagnation (Flipping too late/never).** If the gate remains positive, the agent learns to "hack" local rewards (e.g., infinite loops) without solving the task.
>     *   *Evidence:* Our **"w/o Gating Mechanism"** ablation simulates this. The agent treats valid actions in failures as positive, causing a sharp performance drop (Table 2). This confirms the gate is essential to prevent this failure mode.
> *   **Risk B: Early Collapse (Flipping too early).** If valid actions are penalized before the agent masters syntax, the policy may collapse.
>     *   *Safeguard:* The threshold condition `if V_batch < theta_V` in Eq. 7 acts as a **safety lock**, strictly forbidding sign flipping until action validity matures. This ensures stability in the critical early phase.
> *   **Risk C: The "Repetition" Loop (Degenerate Policy).** The agent discovers a single valid action and repeats it indefinitely to accumulate local rewards (e.g., repeatedly checking the same object).
> *   *Evidence:* In the **"w/o Repetition Penalty"** ablation, the agent collapses into infinite loops. The success rate plummets to near zero, characterizing the severity of this failure mode.
>
>
> **3. Robustness via Adaptive & Cross-Domain Consistency**
>
> Finally, we provide evidence against sensitivity:
> *   **Adaptive, Not Static:** The schedule is **performance-dependent** (based on $C_{batch}, V_{batch}$, $C_{batch}$ is the mean task completion rate, $V_{batch}$ is the mean action validity rate ), not step-dependent. It forms a closed-loop system: if learning slows, the curriculum pauses; if it speeds up, the curriculum accelerates. This naturally minimizes sensitivity to hyperparameters compared to fixed annealing.
> *   **Cross-Domain Consistency:** We applied the **same core gating logic and strategy** across two vastly different benchmarks: **ALFWorld** (text game) and **AppWorld** (coding/API). MGR achieved SOTA on *both*, proving that the method relies on the **general principle of sample balancing** rather than being over-fitted to specific parameters.
> *   **Empirical Stability:** As shown in **Figure 3**, MGR (red curve) exhibits monotonic improvement with **lower variance** than baselines, empirically proving that the gating mechanism enhances rather than disrupts stability.
> *   **Insensitivity:** Those gating parameters act as **loose boundaries** rather than precise values. Any configuration that successfully prevents extreme class imbalance yields similar stable performance. They are akin to setting a "target temperature" for a thermostat; the exact curve matters less than the final equilibrium.

---

> ### Author Response · Authors · 2025-11-24
> **Rebuttal Part3: Response to Weakness 4, Question 1, Question 2**
>
> ---
>
> ### **Response to Weakness 4: Sample Efficiency and Exploration Analysis**
>
> We thank the reviewer for highlighting these critical aspects of RL performance. We respectfully point out that the **Training Dynamics analysis in Section 4.5 and Figure 3** provides direct empirical evidence addressing both sample efficiency and exploration quality.
>
> **1. Sample Efficiency (Evidence from Figure 3)**
>
> MGR demonstrates significantly superior sample efficiency compared to baselines. As shown in the **Task Success Rate curves (Figure 3, Top)**:
> *   **Faster Convergence:** MGR (Red curve) initiates learning much earlier and better.
> *   **Data Efficiency:** This indicates that MGR requires fewer interaction trajectories to attain a capable policy. The decoupling mechanism extracts a clean learning signal from every step (even in failed trajectories), effectively utilizing data that baselines would discard or misuse.
>
> **2. Exploration Analysis**
>
> The reviewer raises a valid concern about exploration. In the context of LLM agents, we argue that the primary bottleneck is not "lack of randomness," but **"ineffective exploration"**: the agent wasting samples on syntactically invalid or hallucinatory actions (e.g., calling non-existent APIs).
>
> *   **Pruning the Search Space:** MGR does not inhibit *strategic* exploration; rather, it inhibits *syntactic* errors. By swiftly enforcing action validity (as evidenced by the rapid rise in **Action Success Rate in Figure 3, Bottom**), MGR effectively **prunes the invalid action space**.
> *   **Enabling Semantic Exploration:** This acts as a foundation for exploration. Once the agent learns *how* to execute actions (Action Validity), it can focus its exploration budget on *what* actions to execute to solve the task. The fact that MGR significantly outperforms baselines on **AppWorld (Hard)** and **ALFWorld (L2)** tasks requiring long, novel sequences of reasoning, which confirms that our method fosters deeper and more effective strategic exploration.
>
> In the revision, we will add a specific paragraph in Section 4.5 to explicitly discuss these quantitative gains in sample efficiency.
>
>
> ---
>
> ### **Response to Question 1: Stability of the Stochastic Gate**
>
> We confirm that the stochastic gate does **not** destabilize learning. As detailed in our **Response to Weakness 3**, the gating mechanism is specifically architected to ensure stability, particularly in the early stages.
>
> **Empirical Evidence (Figure 3)**
>
> The strongest proof is the **Training Dynamics in Figure 3**. The MGR learning curves (Red/Orange) are remarkably smooth and monotonic, whereas baselines (Loop/GRPO) exhibit significant oscillation. This empirical evidence confirms that by resolving the conflicting signals of the additive paradigm, MGR actually **reduces** training variance.
>
> ---
> ### **Response to Question 2: Synergy with Learned Reward Models**
>
> **1. MGR is orthogonal and highly synergistic** with PRMs (e.g., CAPO, RLAIF). The gating mechanism remains essential and non-redundant for two reasons:
>
> 1.  **"Fact" vs. "Estimation":** Learned PRMs provide a **soft semantic estimation** (e.g., "Is this reasoning logical?"), whereas MGR enforces a **hard factual constraint** derived from environment execution (e.g., "Does this code run?").
> 2.  **The "Hallucination Filter":** Even strong PRMs can hallucinate, incorrectly rewarding plausible-looking but invalid actions (e.g., calling non-existent APIs). MGR acts as a **ground-truth, action-level verifier**: it overrides the PRM's score to strictly penalize execution failures. This ensures that high semantic rewards are only attributed to actions that are functionally valid.
>
> In summary, applying MGR on top of PRMs would likely yield the "best of both worlds": the strategic guidance of teacher models and the grounded robustness of execution feedback.
>
>
> **2. MGR as a Fusion Logic for Learned Signals (Beyond Execution)**
>
> MGR can be viewed as a **signal-agnostic framework**, not merely a syntax checker. The equation $R_{\text{final}} = R_{\text{local}} \times R_{\text{global}}$ defines a *fusion logic*, where the source of $R_{\text{local}}$ is interchangeable. In domains lacking action-level signal (e.g., creative writing, summarization, or safety alignment), a **Learned Reward Model (RM)** can serve as the provider of the $R_{\text{local}}$ signal. Thus, MGR can be applied whether the dense signal comes from a compiler (as in our experiments) or a neural network (in broader RLAIF contexts).
>
> ---

---

> ### Author Response · Authors · 2025-11-24
> **Rebuttal Part4: Response to Question 3**
>
> ---
>
> ### **Response to Question 3: Evidence of Gradient Signal Quality**
>
> We thank the reviewer for this deep question. While we did not directly plot gradient norms, we provide both a theoretical guarantee and strong quantitative proxies demonstrating that MGR significantly improves the quality (specifically, the **directional correctness**) of the gradient signal.
>
> **1. Theoretical Guarantee: Elimination of "False Positive" Gradients**
>
> In the additive paradigm, an invalid action $a_{\text{invalid}}$ in a successful trajectory may receive a positive reward $R>0$. This generates a **False Positive Gradient**, pushing the policy to increase the probability of syntax errors. This is mathematically incorrect noise.
> MGR theoretically guarantees that $R(a_{\text{invalid}}) < 0$ always. By strictly eliminating these "False Positive" updates, MGR ensures that the gradient direction **consistently points towards validity**, drastically increasing the Signal-to-Noise Ratio (SNR) compared to the additive baseline.
>
> **2. Quantitative Proxy: Convergence Efficiency (Figure 3)**
>
> In optimization, **convergence speed is a direct proxy for gradient quality**. A higher quality gradient leads to faster traversal of the loss landscape.
> *   **Quantitative Comparison:** As shown in Figure 3 (Top), MGR reaches a 40% task success rate in approximately **40 steps**. In contrast, the GRPO baseline, plagued by noisy additive signals, struggles to reach 30% even after **90 steps**.
> *   **Conclusion:** This **2x+ acceleration** in convergence provides compelling quantitative evidence that MGR provides a cleaner, more informative gradient signal that effectively guides the optimizer, whereas baselines waste samples correcting the noise introduced by credit misassignment.
>
> **3. Ablation Support**
> Furthermore, our ablation study (Table 2) shows that even without a Critic, MGR ("w/o Critic") still outperforms baselines. This suggests that the **raw reward signal** generated by MGR is of sufficiently high quality (low variance, high bias-correctness) to drive learning effectively on its own.

---

### Author Response · Authors · 2025-11-27
**Updates in the Revised PDF**

We have uploaded a revised PDF with changes highlighted in **red**. The key updates are:

*   **Introduction (Section 1):** Refined the text to improve readability.
*   **Method (Section 3):** Added a formal definition for the validity signal ($v_t$) and expanded the motivation for the dynamic gating mechanism using a Signal-to-Noise Ratio (SNR) perspective.
*   **Experiments (Section 4.6 & App. A.6):** Included new sensitivity analyses on hyperparameters ($\beta, \gamma, \alpha$) to demonstrate the robustness of our method.

We thank the reviewers for their suggestions which helped improve the paper's clarity and rigor.

---

### Author Response · Authors · 2025-12-01
**Summary of Rebuttal and Contributions**

---

# Summary of Rebuttal and Contributions

**To the Area Chair:**

---

We understand the significant workload imposed by the recent restructuring of the review process. We appreciate your time and effort in evaluating our work.

We are writing to provide a concise roadmap of our rebuttal, highlighting how we addressed the reviewers' concerns and the consensus reached before the process was halted.

---

### 1. Status of the Review Process
During the rebuttal phase, we engaged in detailed discussions with the reviewers. Notably:
*   **Reviewer tWSq** explicitly stated that their concerns were **solved** by the additional experiments and clarifications and **raised their score**.
*   **Reviewer Ao4r** also confirmed that their concerns were solved and maintained their **positive assessment**.

---

### 2. Synthesis of Key Concerns & Resolutions
The primary concerns centered on three themes. We addressed them as follows:

**A. Novelty: Incremental vs. Foundational**
*   **Concern:** Does MGR just have a incremental novelty?
*   **Response:** We clarified that MGR's novelty lies in identifying and resolving a **fundamental paradigm mismatch** in LLM's agentic RL training methods.

**B. "Heuristics" & Generalization**
*   **Concern:** Reliance on "heuristics" validity checks and the stability of the gating mechanism.
*   **Response:**
    *   **On Validity Check:** We clarified that our checks rely on **universal execution feedback** (Ground Truth) rather than narrow heuristics (specific task logic).
    *   **On Gating:** We demonstrated that the gating mechanism acts as a necessary **implicit curriculum** (Syntax $\rightarrow$ Strategy) that stabilizes training and prevents "reward hacking," rather than introducing bias.

**C. Hyperparameters' Robustness**
*   **Concern:** Sensitivity to ($\alpha, \beta$) and other hyperparameters of gating schedules.
*   **Response:** We provided strong empirical evidence of robustness:
    *   **Zero-Tuning Transfer:** We used the exact same core hyperparameters for both ALFWorld and AppWorld.
    *   **Sensitivity Analysis:** We added ablation experiments, which shows the robustness of key parameters of MGR.
    *   **Error Bounds:** We added multi-seed experiments (Mean $\pm$ Std) confirming our gains are statistically significant over baselines.

---

### 3. Core Contributions & Impact
Our work identifies the "Dual Challenge" of Agentic Learning, which previous method cannot solve simultaneously: **Validity** (Instruction Following/Executability) and **Effectiveness** (Strategic Planning). By demonstrating **SOTA performance**, MGR offers a universally applicable framework for training reliable LLM agents in complex, long-horizon environments.

---

We believe the rebuttal has solidified these contributions and hope this summary assists in your final decision.


**Sincerely,**

**The Authors**

---

### Meta-Review · Area_Chair_Zts4 · 2026-01-07

**Summary:**

The paper proposes Multiplicative Gated Rewards (MGR) for LLM agents, decoupling "local action validity" from "global effectiveness" using a multiplicative rather than additive reward structure.

## Strengths

-  Effectively addresses the "invalid action" problem in LLM agent training.
-  Strong empirical performance and improved convergence on ALFWorld and AppWorld.

## Weaknesses

- The innovation is primarily an empirical heuristic (multiplicative vs. additive rewards) rather than a fundamental algorithmic or theoretical breakthrough.
- High dependency on environment-specific feedback for validity signals, limiting generalizability.

## Reviewer Consensus
While reviewers appreciate the solid experimental results, there is a consensus that the methodological contribution is incremental and more akin to reward engineering.

## Final Decision & Justification
**Decision: Reject**

**Justification:** The submission provides a practical engineering solution with good results. However, it lacks the depth of algorithmic innovation expected for ICLR. The approach is largely heuristic and its broader applicability to unstructured domains remains unproven.

**Reviewer Concerns:**

### Concerns Addressed in Rebuttal

- **Hyperparameter Robustness**: The authors successfully demonstrated that the core parameters (e.g., ) are robust through zero-tuning transfer across heterogenous domains (ALFWorld and AppWorld).


- **Statistical Significance**: Concerns regarding uncertainty and error bounds were mitigated by providing multi-seed experiment results , confirming the gains are statistically significant.

- **Formal Definitions**: The authors addressed the request for clarity by providing a rigorous mathematical definition of the local validity reward  based on syntax and runtime error patterns.

- **Training Stability**: Evidence from training curves effectively showed that the multiplicative formulation reduces oscillation and improves convergence speed compared to additive baselines.

### Outstanding Concerns

- **Incremental Novelty**: Despite the authors' justification, the concern remains that the method is a principled reweighting/masking heuristic rather than a foundational advancement in credit assignment theory.

- **Generalizability to Semantic Tasks**: While the authors argued for "conceptual portability," the method still relies heavily on explicit execution feedback (e.g., error strings, API rejections). Its effectiveness in open-ended or purely semantic tasks without clear "validity" signals remains unproven.

- **Heuristic Nature of Gating**: The batch-stat-driven sign-flip mechanism (gating schedule) is still viewed as a heuristic design lacking a deep theoretical characterization of its failure modes.

**Reviewer Scores:**

### Score Changes

- **Reviewer tWSq**: Increased their score after the authors provided additional experiments and clarifications. They noted their concerns were addressed.

### No Score Change

- **Reviewer BvSX**: Kept their score at **4** (marginally below acceptance). While acknowledging the method's strengths, they maintained that the novelty is incremental.

- **Reviewer LUMB**: Kept their score at **6** (marginally above acceptance). They remained concerned about hyperparameter sensitivity and the heuristic nature of the validity signals.

- **Reviewer Ao4r**: Maintained their positive assessment **6**.

---

### Decision · Program_Chairs · 2026-01-26

Reject